# Cell Death in Liver Diseases: A Review

**DOI:** 10.3390/ijms21249682

**Published:** 2020-12-18

**Authors:** Layla Shojaie, Andrea Iorga, Lily Dara

**Affiliations:** 1Division of Gastrointestinal & Liver Diseases, Department of Medicine, Keck School of Medicine, University of Southern California, Los Angeles, CA 90033, USA; layla.shojaie@usc.edu (L.S.); andrea.iorga@med.usc.edu (A.I.); 2Research Center for Liver Disease, Keck School of Medicine, University of Southern California, Los Angeles, CA 90033, USA

**Keywords:** hepatocytes, apoptosis, necrosis, necroptosis, pyroptosis, ferroptosis, NASH, hepatotoxicity, RIPK1, RIPK3

## Abstract

Regulated cell death (RCD) is pivotal in directing the severity and outcome of liver injury. Hepatocyte cell death is a critical event in the progression of liver disease due to resultant inflammation leading to fibrosis. Apoptosis, necrosis, necroptosis, autophagy, and recently, pyroptosis and ferroptosis, have all been investigated in the pathogenesis of various liver diseases. These cell death subroutines display distinct features, while sharing many similar characteristics with considerable overlap and crosstalk. Multiple types of cell death modes can likely coexist, and the death of different liver cell populations may contribute to liver injury in each type of disease. This review addresses the known signaling cascades in each cell death pathway and its implications in liver disease. In this review, we describe the common findings in each disease model, as well as the controversies and the limitations of current data with a particular focus on cell death-related research in humans and in rodent models of alcoholic liver disease, non-alcoholic fatty liver disease and steatohepatitis (NASH/NAFLD), acetaminophen (APAP)-induced hepatotoxicity, autoimmune hepatitis, cholestatic liver disease, and viral hepatitis.

## 1. Introduction

### 1.1. Apoptosis

Apoptosis, derived from the Greek word for falling leaves, is the most comprehensively studied and described form of programmed cell death. Apoptosis can be triggered either intrinsically or extrinsically [1]. Both pathways lead to the activation of the executioner caspases 3 and 7 (CASP3, CASP7), resulting in proteolysis, nuclear fragmentation, and apoptotic cell death. Cellular injury such as DNA damage, starvation, or oxidative stress can activate the intrinsic apoptotic pathway [1]. The intrinsic pathway of apoptosis is defined by the nomenclature committee on cell death (NCCD) as a form of regulated cell death that is activated by perturbations in the intracellular and extracellular microenvironment characterized by mitochondrial outer membrane pore formation (MOMP) and precipitated by executioner caspases [1]. MOMP is controlled by the balance between the pro-apoptotic and anti-apoptotic members of the B cell lymphoma-2 (Bcl-2) family of regulator proteins [2]. In apoptosis, the activator Bcl-2 protein, BH3-interacting domain death agonist (BID), undergoes post-transcriptional modification and cleavage, forming tBID. Then, tBid translocates to the mitochondria and interacts with the mitochondrial pool of proapoptotic Bcl-2 members, BCL2-associated X apoptosis regulator (BAX), and/or BCL2 antagonist/killer (BAK) [1]. This results in conformational changes, leading to MOMP and mitochondrial release of apoptotic constituents such as cytochrome *c* and second mitochondria-derived activator of caspase/direct inhibitor of apoptosis binding protein with low pl (Smac)/DIABLO. Cytochrome *c* binds apoptotic peptidase activating factor 1 (APAF1) and pro-caspase 9 (pro-CASP9), forming the apoptosome [3]. The apoptosome advances the self-activation of CASP9, ultimately resulting in activation of executioner caspases (CASP3 and CASP7), causing cell death (Figure 1A) [1].

The extrinsic pathway of apoptosis is initiated by perturbations of the extracellular microenvironment and mostly driven by stimulation of transmembrane receptors such as death receptors (DRs) or pattern recognition receptors (PRRs) [1]. Ligation to DRs, such as Tumor Necrosis Factor Receptor (TNFR), FAS, tumor necrosis factor-related apoptosis-inducing ligand receptor (TRAIL), or PRRs such as Toll-like receptors (TLRs), results in the assembly of a multiprotein complex called the Death-inducing signaling complex (DISC) or complex 1, which regulates caspase 8 (CASP8) activity and downstream interactions [1]. Multiple proteins form complex 1, including receptor-interacting protein kinase-1 (RIPK1), cellular inhibitor of apoptosis 1 and 2 (cIAP1 and 2), TNF receptor-associated factors 2 or 5 (TRAF2 or TRAF5), and the adaptor TNFR-associated death domain (TRADD). It is well known that DR ligation and in particular TNFR ligation does not always result in cell death. RIPK1 has a direct influence on the outcome of TNFR activation, resulting in pro-survival vs. pro-death pathways for the affected cell depending on its post-translational modifications [4,5]. The E3 ubiquitin ligases cellular inhibitor of apoptosis (cIAPs) catalyze the K63 polyubiquitination of RIPK1, forming a platform that facilitates the activation of the Nuclear factor-κΒ (NFκΒ) pathway through transforming growth factor-activated kinase-1 (TAK1) and TAK-1 binding proteins (TAB2 and 3) [6]. NFκΒ activation subsequently leads to the transcription of pro-survival genes that prevent cell death [6,7]. On the other hand, if RIPK1 is deubiquitinated, it is released from complex 1 to form complex 2, where it binds to FADD and CASP8 [6,7]. In lymphocytes and other Type I cells, CASP8 cleavage leads to apoptotic cell death by the subsequent activation of executioner CASP3 and CASP7 [8]. However, in type II cells such as hepatocytes, the formation of complex 2 is constitutively blocked by x-linked inhibitor of apoptosis (XIAP), NFκΒ target genes such as TNF alpha induced protein 3 (TNFAIP3, also known as A20), and Fas-associating protein with death domain-like interleukin-1 -converting enzyme (FLICE) inhibitory protein (c-FLIP) [1,9]. c-FLIP, a catalytically inactive close relative of CASP8, exists in a long form (cFLIP_L_) and short form (c-FLIP_S_) and is a regulator of CASP8 activation. cFLIP_L_ heterodimer binds to CASP8 (CASP8-cFLIP_L_) promoting CASP8 oligomer assembly, while the short heterodimer cFLIP_S_ (CASP8-cFLIP_S_) blocks CASP8 enzymatic activity [10]. It is important to note that cFLAR, the gene encoding cFLIP, is under the transcriptional control of NFκΒ (Figure 1A) [1]. In type II cells, extrinsic apoptosis requires the cleavage of BID to tBID and its translocation to mitochondria, resulting in MOMP-driven apoptosome formation and cell death as described above. Therefore, in hepatocytes, the amplification of the death signal and mitochondrial engagement is necessary for DR-induced cell death [9].

### 1.2. Mitochondrial Permeability Transition-Driven Necrosis

Necrosis is a form of cell death characterized by cell swelling and loss of plasma membrane integrity [1]. Mitochondrial Permeability Transition (MPT)-driven necrosis is a form of regulated cell death (RCD) initiated by toxins or perturbations in the cellular microenvironment due to oxidative stress, which results in the abrupt loss of mitochondrial membrane potential and cell membrane rupture [1]. Morphologically, necrosis contrasts with apoptosis, such that necrosis results in nuclear condensation, cell shrinkage, and blebbing, although secondary necrosis following apoptosis has been described [11]. Following necrosis, cellular contents, including cytotoxic and pro-inflammatory factors such as danger-associated molecular patterns (DAMPs), are released into the extracellular space [12]. The exact mechanisms leading to MPT have not been fully elucidated; however, this pore is proposed to be composed of dimers of the ATP synthase complex, which can be opened by interacting with the mitochondrial protein cyclophilin D (CypD) [13]. CypD inhibitors, such as cyclosporin A (CsA), or knockdown and knockout (KO) of CypD have shown promise in preventing necrosis in animal models of oxidative stress such as renal, cardiac, liver, and neurologic ischemia reperfusion, neurodegeneration, cancer, as well as acetaminophen (APAP) toxicity and hepatic steatosis [14,15,16]. While the role of CypD in MPT is experimentally evident, its translational relevance and application to human disease has been called into question due to negative results with trials of cyclosporin A (CsA) preventing cardiac myocyte death in humans undergoing cardiac catheterization [17]. Therefore, much remains unknown regarding the precise mechanism and signaling events leading to regulated necrosis from MPT.

### 1.3. Necroptosis

The best known and most extensively studied pathway of regulated necrosis is necroptosis, which was initially described when a shift in the cell death mode from apoptosis to necrosis was observed upon TNFR stimulation with TNF in the presence of caspase inhibitors [18]. Necroptosis is initiated by DRs, PRRs such as TLRs, or intracellular sensors such as Z-DNA binding protein 1 (ZBP1/DAI). This RCD pathway occurs with caspase inhibition and necessitates the proteins RIPK1, RIPK3, and the pseudokinase mixed lineage domain-like (MLKL) [19,20,21,22,23]. Complex 1 is created following DR ligand binding. In several cell types, when CASP8 is inhibited, deubiquitinated RIPK1 does not associate with complex 2 but alternatively binds to RIPK3 through the shared RIP homology interaction motif (RHIM) [22]. Then, the necrosome complex is formed, which is followed by the RIPK3 recruitment of MLKL. RIPK3 phospho-activates MLKL (forming pMLKL), driving its translocation to the plasma membrane, where it oligomerizes and induces pore opening and necroptotic cell death [22,24]. Necroptotic cell death is kept in check by CASP8-mediated cleavage of RIPK1 and RIPK3, such that in the presence of CASP8, the cell death mode defaults to apoptosis [25,26]. Accordingly, an inhibition of CASP8 is imperative for necroptosis to occur, raising the question regarding the contribution of necroptosis to the pathology of human disease conditions where caspases are intact and not inhibited (Figure 1B) [27]. The kinase activity of RIPK1 is necessary for necroptosis and RIPK1-dependent apoptosis to proceed but is nonessential for its pro-survival function [28]. The kinase activity of RIPK3, which activates MLKL, is similarly essential for necroptosis [23]. The role of the RIPK proteins, MLKL and necroptosis in liver disease and liver cell death has garnered much controversy in the past few years [27,29]. Complicating matters further are the intricacies and the various non-necroptotic functions of these proteins and in particular the role of RIPK1 and RIPK3 in apoptosis and inflammation [23,29]. Under basal conditions, RIPK3 is undetectable in liver cells [30], and its induction is controversial; therefore, the role of necroptosis in hepatocyte cell death in liver diseases is under extensive investigation [27]. However, RIPK3 has been detected in non-parenchymal cells (NPCs), and MLKL is expressed in all liver cell types. MLKL also has non-necroptosis functions; for example, it has been shown to inhibit autophagy and to play an important role in vesicle trafficking. [31,32]. Therefore, the possibility of necroptosis contributing as a mode cell death in these cell types in the liver remains an intriguing idea that warrants further investigation [29,33,34,35,36].

### 1.4. Autophagy and Autophagy-Dependent Cell Death

Autophagy is an intracellular waste degradation pathway that occurs through the formation of the autophagosome. Autophagy plays a pivotal role in cell survival [37]. Autophagy is activated in response to cellular stress, mediating protective rather than cytotoxic effects [1]. Defects in the autophagic machinery have been associated with various pathological settings. Blocking autophagy commonly expedites the stress response of cells during pathological conditions [1]. In fact, the concept of autophagic cell death is thought to be rather controversial. Components of the autophagy machinery closely interact with the apoptotic machinery, and in certain contexts and specific models, they have been associated with the promotion of cell death. Much of the discussion on autophagy-dependent cell death occurs during development and has been studied in Drosophila [1]. However, certain mammalian examples exist. For example, the deletion of autophagy-related gene 7 (*Atg*7) prevents neurotoxicity in mouse models of hypoxia–ischemia, or the blockade of Atg5 prevents the neurotoxicity of cocaine [1]. In liver disease models where autophagy plays a prominent role, such as alcoholic steatohepatitis, autophagy has mostly proven to be a protective rather than death-inducing mechanism. During autophagy, double membrane autophagosomes fuse with the lysosome to form autolysosomes within which autophagic cargo is degraded by hydrolases [1]. The KO of *ATG* genes expedites cell death in most models [38]. However, in particular instances, autophagy eventually leads to RCD [1]. The term “Autophagic cell death” is advised to be used when markers of autophagy and autophagic degradation substrates are elevated. Most importantly, when cell death can be prevented with autophagy inhibition [38]. The interaction of autophagy with apoptotic cell death, necroptosis, and ferroptosis has been extensively studied in various organs, including the liver [39,40,41].

### 1.5. Pyroptosis

Pyroptosis is a profoundly inflammatory mode of RCD related to the innate immune system [42]. It has evolved to remove intracellular pathogens and has a distinct morphology associated with cell bursting. The canonical pathway of pyroptosis occurs when inflammasome sensors, NOD-like receptor family, pyrin domain-containing-1 and 3 (NLRP1, NLRP3), or absent in melanoma-2 (AIM2) are stimulated by pathogens, pathogen-associated molecular patterns (PAMPs), and DAMPs and recruit CASP1 to activate Gasdermin D (GSDMD), which forms a pore in the plasma membrane [1]. In the non-canonical pathway, cytosolic LPS and PAMPs stimulate CASP4, 5, and 11 directly, which in turn cleave GSDMD. Then, activated GSDMD, the main conduit of pyroptosis, binds membrane phospholipids and initiates pore formation, resulting in cell death (Figure 1C) [43,44,45]. The contribution of pyroptosis to liver disease and in particular Non-Alcoholic Steatohepatitis (NASH) is the topic of intense research in Hepatology [46,47]. Unrepressed NLRP3 activation has been shown to result in shortened survival, severe liver inflammation, and hepatic stellate cell (HSC) activation, leading to collagen deposition and liver fibrosis [48].

### 1.6. Ferroptosis

Ferroptosis is another form of RCD characterized by glutathione (GSH) depletion and severe iron-dependent lipid peroxidation in a Fenton-like manner, resulting in reactive oxygen species (ROS) production and cell death [49,50]. Ferroptosis has been shown to contribute to cancer-related cell death, as well as ischemia reperfusion injury, neurologic diseases, and acute kidney injury [51]. This form of RCD occurs independently of caspases, necroptosis, autophagy, and pyroptosis. However, it does have subcellular elements reminiscent of necrosis and could be associated with a release of DAMPs [1]. The presence of miniature mitochondria with compressed mitochondrial densities with vanished cristae are morphological hallmarks of ferroptosis [52,53]. The GSH-dependent enzyme glutathione peroxidase-4 (GPX4) is the main endogenous inhibitor of ferroptosis due to its ability to limit lipid peroxidation by reducing lipid peroxides to alcohols [1,54]. The inhibition of GPX4 activity can lead to the accumulation of lipid peroxides, which is a marker of ferroptosis (Figure 1D). Many of the signaling events and effectors of ferroptosis have been elucidated by performing experiments using specific activators (such as RSL3 and erastin) and inhibitors (such as ferrostatins) of ferroptosis. For example, the commonly used agent, Erastin, activates ferroptosis by inhibiting *System X_c_^−^*, the cystine/glutamate antiporter system, which then decreases intracellular cysteine and limits GSH synthesis [51]. The role of ferroptosis has been newly investigated in various models of liver disease and will be further discussed in our review.

### 1.7. Other Modes of Cell Death

Other forms of cell death such as NETosis, parthanatos, entotic cell death, and lysosome-dependent cell death have been described in the literature, which are not covered here due to the lack of robust data on these cell death subroutines in liver diseases [1].

## 2. Cell Death in Alcoholic Liver Disease (ALD)

### 2.1. Apoptosis in ALD

Apoptotic, necrotic, autophagic, pyroptotic, and ferroptotic cell death pathways have all been implicated in the pathogenesis of alcoholic liver diseases (ALD) to some extent [55,56,57]. Apoptosis is the most studied form of RCD in ALD. In animal models and in vitro, alcohol induces metabolic, toxic, and inflammatory insult, leading to mitochondrial dysfunction, the generation of ROS, Bax translocation to mitochondria, cytochrome *c* release, and caspase activation [58,59,60]. The metabolism of alcohol plays a significant role in alcohol-induced mitochondrial and endoplasmic reticulum (ER) stress, resulting in apoptotic cell death [55]. Alcohol and its metabolites such as acetaldehyde are highly reactive and can cause increased ROS production and misfolded protein accumulation, triggering ER stress. This ER stress response can lead to mitochondrial stress and CHOP-dependent apoptosis [61,62,63]. Additionally, ER stress triggered by alcohol has been shown to promote the association of interferon regulatory factor 3 (IRF3) with a stimulator of interferon genes (STING), resulting in the phospho-activation of IRF3. Phospho-IRF3 has then been shown to associate with Bax, with the ultimate outcome of hepatocyte apoptosis [64]. In addition to ER and mitochondrial stress, alcohol also raises lysosomal pH, thus leading to lysosomal malfunction. These organelle stress-mediated responses result in the engagement of the intrinsic pathway of apoptosis and inflammation [55,65].

The extrinsic apoptosis pathway ligands, FASL and TNF, have a critical role in the pathogenesis of ALD [66,67]. Acute and chronic alcohol exposure alters intestinal permeability, leading to elevated systemic bacterial products such as lipopolysaccharide (LPS), which in turn result in Kupffer cell (KC) stimulation and TNF production [68]. Increased FAS receptor expression in alcoholic steatohepatitis is more prominent that TNF-R1 [69]. The importance of TNF in mediating alcohol-induced inflammation and cell death has been well characterized. Circulating levels of TNF and TNFR are elevated in patients with alcoholic liver disease and alcoholic steatohepatitis, while treatment with anti TNF antibodies protects against alcoholic liver injury in animal models [60,70]. In liver biopsies of patients with alcoholic hepatitis, increased Terminal deoxynucleotidyl transferase dUTP nick end labeling (TUNEL) staining and CASP3 positive hepatocytes have been detected [69]. This was significantly higher in those with elevated bilirubin (>3 gr/dL) and higher-grade steatohepatitis. Despite the clear association of TNF with alcoholic liver disease, treatment with the anti-TNF antibody infliximab has been proven detrimental in patients with alcoholic steatohepatitis due to increased rates of infection and mortality [71].

The role of CASP8, the apical initiator caspase, remains unclear in ALD. In a study by Hao et al., the KO of CASP8 in hepatocytes (Casp8^Δhepa^) failed to prevent alcohol-induced apoptosis (Lieber–DeCarli model) [72]. Cell death through the engagement of the intrinsic mitochondrial pathway was even more pronounced in these mice [72]. Interestingly, Casp8^Δhepa^ mice fed with alcohol had attenuated steatosis and triglyceride (TG) content compared to floxed controls [72]. In vitro and in vivo studies using a pan-caspase inhibitor have shown considerable attenuation of alcohol-induced hepatocyte apoptosis with no switch to necroptosis and no induction of RIPK1, RIPK3, or MLKL [72,73]. Global BID KO mice treated with a pan-caspase inhibitor displayed less apoptosis and decreased TUNEL-positive hepatocytes when compared to wild-type (WT) mice after chronic alcohol feeding. However, pan-caspase inhibition and Bid deficiency had no effect on transaminases, steatosis, or the expression of inflammatory cytokines [74]. Despite a lack of effect on inflammatory markers, fibrosis was attenuated with pan-caspase inhibition in a model of cotreatment with alcohol and carbon tetra chloride (CCL4) in this study [74]. Interestingly, in another study by the same group, chronic alcohol feeding also led to BID-dependent apoptosis in adipose tissue, which was attenuated by BID KO [75].

### 2.2. Necroptosis in ALD

Necroptosis requires an interplay between RIPK1 and RIPK3, the activation of MLKL (p-MLKL), and its translocation to the cell membrane [1]. Although RIPK3 is not present under basal conditions in hepatocytes following alcohol binge, the induction of RIPK3 in whole liver lysates and on immunostaining has been observed [30,76]. It is unclear whether the source of this accumulating RIPK3 is from the hepatocyte or from the non-parenchymal compartment. RIPK3 global KO mice treated with alcohol had less steatosis, inflammation, and liver injury compared to WT controls [76]. However, daily in vivo treatment with the RIPK1 inhibitor (Necrostatin-1, Nec-1) was not protective against alcohol in mice [76]. In a Gao-binge model of alcohol, Wang and colleagues have also explored the contribution of RIPK3 to ALD [77]. RIPK3 global KO mice had decreased serum transaminases and steatosis, but there was no difference in hepatitis and neutrophil infiltration compared to WT mice. There was no transcriptional induction of RIPK3 in the Gao-Binge mice, and the increased RIPK3 protein was due to decreased proteasomal turnover. Alcohol decreased hepatic proteasome function. In support of this, pharmacological and genetic inhibition of the proteasome resulted in the accumulation of RIPK3 in mouse livers [77]. Interestingly, treatment with 7-Cl-O-Nec-1 (7-Nec-1), a specific inhibitor of RIPK1, reduced hepatic inflammation, neutrophil infiltration, and NFκB nuclear translocation, but it failed to protect against steatosis and liver injury induced by alcohol [77]. There are no studies examining the impact of MLKL deletion in mouse models with ALD. While the accumulation of RIPK3 could promote more MLKL phosphorylation and necroptosis, until such models are explored, the contribution of necroptotic cell death to ALD remains unconfirmed.

### 2.3. Autophagy in ALD

Expanding evidence suggests that autophagy plays an important role in ALD [41,78,79]. Autophagy and apoptotic cell death can regulate one another. Autophagy functions to remove and recycle excess proteins and damaged organelles and loss of autophagy in hepatocytes, for example by ATG5 KO, results in liver cell death, inflammation, and fibrosis [80]. As alcohol-induced ER stress and proteosome inhibition have been shown to activate autophagy [78], it has been proposed that autophagy activation and the removal of aggregated proteins and damaged organelles can attenuate alcohol-induced ER stress and subsequent cell death [55,81,82]. Specifically, since mitochondria initiate apoptosis and release ROS, the mitophagy of damaged and stressed mitochondria is beneficial for cell survival due to the removal of cellular debris, lipid droplets, and protein aggregates [78,83].

Acetaldehyde, an oxidative stress inducing metabolite of ethanol, is an activator of autophagy [82]. Since the resultant ROS generation from ethanol and its metabolites also activates other cellular stress responses that can lead to cell death, parsing out the exact contribution of autophagy during ethanol injury is challenging. However, it is well accepted that the activation of autophagy is protective against alcohol-induced hepatocyte injury and death [82]. Ethanol induces autophagy through oxidative stress mechanisms involving the activation of (5′ AMP-activated kinase) AMPK and the suppression of (mammalian target of rapamycin complex 1) mTORC1 [79,82]. Indeed, the activation of autophagy by the mTORC1 inhibitor, rapamycin, has been shown to reduce alcohol induced liver injury, and inhibiting autophagy using chloroquine has been shown to be detrimental [79]. Although acute alcohol ingestion activates autophagy, when given chronically and at high doses, alcohol can suppress autophagy [79,82]. As expected, autophagy suppression leads to increased liver injury, ALT elevations, and hepatocyte death [79]. We now know that whether alcohol exposure leads to increased or decreased autophagy depends on several factors. The amount of alcohol in the diet, the duration of alcohol exposure, and the alcohol delivery method can all influence the effect of ethanol on autophagy [82]. While acute ethanol exposure increases autophagy [78], chronic alcohol exposure has conflicting results. Chronic low-dose alcohol exposure can lead to increased autophagy; however, the suppression of autophagy markers has been noted with high doses of ethanol [79,82]. Importantly, in both chronic and acute ethanol exposure, autophagy suppression consistently worsened liver injury, while the promotion of autophagy improved toxicity [79].

The phosphatase and Tensin homolog (PTEN)-induced putative kinase 1 (Pink1)–Parkin axis plays a crucial role in mitophagy in mammalian cells [84]. Parkin is an E3 ubiquitin ligase encoded by the Park2 gene and implicated in Alzheimer’s disease [85]. Parkin-mediated mitophagy is necessary for liver homeostasis and response to alcohol, as Parkin KO mice treated with alcohol had more mitochondrial damage, steatosis, and liver injury compared to WT controls [85,86]. In addition to decreased mitophagy, Parkin KO mice had less β-oxidation as well as decreased mitochondrial respiration and cytochrome *c* oxidase activity after acute alcohol treatment compared to WT mice [86].

Sorting nexin (SNX)-10 has a regulatory function on chaperone-mediated autophagy (CMA). You et al. found that SNX10 deficiency both in vivo and in vitro upregulated lysosome-associated membrane protein type 2A (LAMP-2A), which then lead to CMA activation and alleviated ethanol-induced liver injury and steatosis [87]. SNX10 knockout also inhibited Cathepsin A (CTSA) maturation, leading to increased LAP-2A stability, promoting CMA, and thus alleviating alcohol toxicity [87].

As we have outlined, the process of autophagy is a mainly cytoprotective mechanism in alcoholic liver disease. However, there is a close relationship between autophagy and apoptosis, and the crosstalk between these pathways during ALD will determine cell fate [88].

### 2.4. Pyroptosis in ALD

Recently, pyroptosis has garnered much attention as a potential cell death subroutine contributing to the pathogenesis of ALD [89]. NLRP3 deficiency prevents inflammation and improves injury by reducing DAMPS, as well as the paracrine interaction of hepatocytes and immune cells [43,89,90]. The specific cell types leading to liver injury by pyroptosis, as well as the involvement and role of hepatocytes, remain to be clarified. Heo et al. suggest that pyroptosis occurs in hepatocytes with alcohol exposure as NLRP3, apoptosis-associated speck-like protein containing a CARD (ASC), and CASP1 were all upregulated with ethanol [91]. In human liver biopsies of patients with alcoholic hepatitis (AH) and in experimental mouse models treated with alcohol, miR-148a levels were decreased [91]. Reduced miR-148a resulted in the overexpression of thioredoxin-interacting protein (TXNIP), which is a protein that can activate the NLRP3 inflammasome and subsequent pyroptosis. However, only macrophage depletion was considered to assess the contribution of other cell population(s) to liver injury in this study [91].

ALD is known to induce an inflammatory milieu with upregulation of pro-inflammatory cytokines. Petrasek and colleagues have demonstrated that alcohol induces IL-1, as well as pro-CASP1 and NLRP3 in mice. Using an IL-1 receptor antagonist, or mice deficient in CASP1 or ASC, the study demonstrated that CASP1-mediated IL1β signaling is necessary for the development of steatosis, inflammation, and injury in ALD [92]. TLR4 signaling is also required for the pathogenesis and development of ALD [93]. TLR4 can activate myeloid differentiation primary response protein (MyD88)-dependent and independent pathways [94]. Using myeloid-specific MyD88-deficient (MyD88LysM-KO) mice and MyD88fl/fl controls fed the Lieber-DeCarli diet, MyD88fl/fl mice developed early alanine aminotransferase (ALT) elevation, inflammation, and steatosis [95]. Cleaved CASP1 and mature IL-1β were present in MyD88fl/fl mice but not in MyD88LysM-KO mice. Interestingly, TUNEL staining revealed no difference in apoptotic cell death [95].

Khanova et al. studied patients with AH and developed a mouse model to mirror AH, consisting of the Western diet combined with chronic intragastric alcohol together with an additional weekly binge dose to mimic human disease [96]. Interestingly, mice on this diet displayed no alterations in CASP1 and IL-1β. However, there was an increase in CASP11 in mice and CASP4 in human liver biopsies from AH patients, but not in chronic ALD and healthy livers [96]. Similar to CASP1, in response to pathogens and PAMPs, CASP11/4 can cleave and activate GSDMD through the non-canonical pyroptosis pathway. Cleaved GSDMD was detected in isolated hepatic macrophages but less clearly in hepatocytes. The overexpression of cleaved GSDMD in hepatocytes resulted in increased cell death [96]. These data suggest that pyroptosis may play a role in the pathogenesis of AH, which is a severe inflammatory disease. Furthermore, this mode of cell death occurred more robustly in macrophages and was executed through the non-canonical rather than the canonical pyroptotic pathway [96]. Additional studies using GSDMD hepatocyte KO mice and macrophage/myeloid conditional KOs are necessary to clearly delineate the contribution of pyroptotic cell death in ALD and AH.

### 2.5. Ferroptosis in ALD

A handful of studies have explored the role of ferroptotic cell death in the pathogenesis of ALD [57,97]. Sirtuin 1 (SIRT1) is a class III histone deacetylase, which is a regulator of lipid metabolism and inflammation [97]. Intestine-specific aberrant liver sirtuin 1 (SIRT1) deficiency may have a protective effect on iron metabolism through increasing glutathione stores, dampening lipid peroxidation, and thereby decreasing genes related to ferroptosis. [98]. A recent study by Zhou et al. conducted with chronic alcohol feeding plus binge on SIRT1 intestinal specific KO mice revealed these mice had improved alcohol-induced iron metabolism, cytokine recruitment, and ultimately less liver injury [98]. In another study by the same group, Lipin-1, another regulator of lipid metabolism, was investigated in mice using the chronic-plus-binge ethanol feeding protocol [57] Adipose-specific lipin-1 overexpressing transgenic mice (*Lpin1*-Tg) showed worsened steatohepatitis, higher aspartate aminotransferase (AST)/ALT levels, extensive iron accumulation, decreased GSH, and impaired ferroptotic gene expression (such as aldo-keto reductase family 1 member C6, glutaminase 2, and solute carrier family 1 (neutral amino acid transporter) member 5) [57]. However, GPX4, which is considered a key enzyme of ferroptosis, was not significantly altered (neither mRNA nor protein levels), suggesting that alternate pathways may initiate ferroptosis in this model [57].

## 3. Cell Death in Nonalcoholic Fatty Liver Disease NASH/NAFLD

Non-alcoholic fatty liver disease (NAFLD) is the most common cause of liver disease in the U.S [99]. Non-Alcoholic Steatohepatitis (NASH), which occurs in 20% of those with NAFLD, is characterized by steatosis accompanied by ballooned hepatocytes (undergoing cell death), Mallory bodies, and inflammation leading to fibrosis. There is strong evidence that hepatocyte cell death drives inflammation and fibrosis in NASH [100]. However, the mode of cell death in this metabolic disorder has been debated. Multiple animal models of fatty liver disease have been developed to study the pathogenesis of NASH, each with varying degrees of inflammation, weight gain, insulin resistance, and cell death. The most widely accepted model of NASH/NAFLD is the high-fat diet (HFD) or Western diet, which comes in different formulations, various fat and cholesterol concentrations, and can be given with the addition of fructose [101]. Another widely used model is the methionine-choline deficient (MCD) diet, which is a diet deficient in the essential nutrients methionine and choline that results in the activation of the integrated stress response, leading to hepatitis and inflammation with a lean body phenotype and no insulin resistance [102]. In addition to diet models, some investigators have used genetically engineered mice such as the leptin deficient *OB/OB* mice or the *db/db* mice that have a natural mutation in the leptin receptor to recapitulate human steatohepatitis [103]. The heterogeneity in these various animal models has resulted in much controversy and conflicting results in studying all aspects of NASH, including the cell death subroutines involved in driving inflammation and fibrosis. More recently, investigators have shied away from genetic models and nutrient deprivation in favor of using WT mice fed high-calorie and high-cholesterol diets that cause obesity, steatohepatitis, and insulin resistance and more closely mimic human pathophysiology [101]. However, heterogeneity remains even among the HFDs used. Furthermore, extra dietary factors such as time of feeding, housing temperature, and the microbiome of the animals can result in confounding results and lack of reproducibility [101]. Therefore, as we discuss the various cell death modes attributed to NASH/NAFLD, we keep in mind the animal model and the diet used, as these become important in the interpretation of findings and can affect the mechanism of injury, inflammation, and cell death subroutine [104].

### 3.1. Apoptosis in NASH/NAFLD

There is strong evidence that apoptotic cell death drives inflammation in NASH [99]. This includes the detection of CASP 3/7 as well as increased TUNEL positivity in liver biopsies of patients with NASH [99]. In accordance, CASP3 and CASP8 KO mice fed an MCD diet have been shown to be protected from apoptosis, and display less inflammatory cytokine signaling, leading to less inflammation and fibrosis [105,106]. Mice fed an HFD demonstrate increased products of lipid peroxidation, apoptosis by TUNEL assay, and increased CASP3 and CASP8 activity as well [107]. The addition of the pan-caspase inhibitor, Emricasan or IDN-6556, significantly attenuated liver injury, inflammation, fibrosis, and apoptosis in mice fed with HFD [107].

In a study be Witek et al. on obese *db/db* mice models fed an MCD diet, four-week treatment with pan-caspase inhibitor VX-166 improved MCD-induced steatosis, but this effect disappeared after eight weeks on MCD [108]. Treatment with VX-166 and inhibition of apoptosis significantly reduced fibrosis; however, hepatocyte ballooning, NAFLD activity score, and inflammation were not decreased [108]. However, in a more recent study, Anstee et al. failed to detect an improvement in steatosis with VX-166 in MCD and HFD-fed mice [109]. Caspase inhibition resulted in improved ALT levels, decreased histological inflammation, and oxidative stress, and it led to less apoptotic cell death, particularly in the MCD-fed diet mice [109].

Despite the salutary effects of pan-caspase inhibition in animal models and evidence of decreased inflammation and apoptosis activation in early small studies [110], pan-caspase inhibitors have not been successful in large randomized clinical trials (RCT) of NASH [111]. In patients with fibrosis grade 1 to 3, Emricasan resulted in more hepatocyte ballooning and fibrosis while modestly improving ALT levels and CASP3 activity, while it failed to demonstrate an improvement in liver histology [112]. In compensated cirrhotics, Emricasan failed to improve portal hypertension or clinical outcomes, although compensated patients with higher baseline HVPG displayed evidence of a small treatment effect [113]. These recent findings have highlighted the intricate interplay between the various forms of cell death. Inhibiting apoptosis may activate alternate cell death subroutines such as necroptosis, resulting in more hepatocyte ballooning and a lack of benefit from Emricasan.

The importance of apoptosis in NASH is further demonstrated by the crosstalk observed between the metabolic sensor adenosine monophosphate (AMP)-activated protein kinase (AMPK) and apoptosis seen in the Amylin diet and choline-deficient HFD (CD-HFD) [114]. Although the role of CASP3 in NASH was known, CASP6 was recently shown to be instrumental in NASH progression and apoptosis [114]. Importantly, CASP6 was activated in livers of humans with NASH. The authors demonstrated that inflammation in NASH leads to the activation of CASP3 and CASP7, which in turn cleave and activate CASP6. Activated CASP6 cleaves Bid, resulting in mitochondrial cytochrome *c* release, which activates CASP3 and 7, leading to a feedforward loop and persistent apoptosis in hepatocytes [114]. In a healthy liver, AMPK is active and phosphorylates proapoptotic CASP6, inhibiting its activation and keeping this loop in check. In these NASH models, AMPK was shown to be suppressed [114]. Zhao and colleagues demonstrated the mechanistic importance of this AMPK axis by employing an AMPK agonist or a CASP6 inhibitor to improve liver transaminases and to decrease apoptosis and liver fibrosis [114]. CASP8 and FADD-like apoptosis regulator (CFLAR)/cFlip have been reported as an essential suppressor of steatohepatitis and its metabolic disorders. Wang et al. report that cFlip can directly interrupt the activation of the mitogen-activated protein kinase (MAPK) apoptosis signal-regulating kinase 1 (ASK1) dimerization, thus inhibiting NASH by blocking ASK1 to cJun-N-terminal Kinase (JNK) signaling [115]. Additionally, the same group has reported that TNFAIP3 can endogenously suppress ASK1 and reduce apoptosis, lipid accumulation, and inflammation [116].

There is a clear link between the unfolded protein response (UPR) sensors that activate the ER stress response and insulin resistance, NASH, and lipotoxicity [117]. Once activated, the ER stress response leads to apoptotic cell death through JNK-mediated Bim activation and inhibition of the pro-survival Bcl2s. Hepatocellular ballooning is one of the characteristics of lipotoxic liver injury. To assess the role of ER stress in ballooning degeneration, Nakagawa et al. injected WT mice fed an HFD with an ER stress elicitor, tunicamycin together with a protein glycosylation inhibitor and detected ballooning degeneration, hepatocyte apoptosis, and ALT rise [118]. Hepatocyte lipotoxicity is a feature of NASH and results in cell and organelle stress, apoptosis, and the release DAMPs. Then, these DAMPs can activate TLRs, leading to a perpetuation in the inflammatory and cell death signals. The interplay between the immune system and hepatocytes contributes to the pathogenesis of NASH as well. In fact, macrophages, KCs, and other WBCs such as natural killer (NK) cells are the main source of TNF in steatohepatitis, promoting inflammation. Roh and colleagues have shown that TLR4 and TLR7 inhibition improve NASH. In this model, TLR-7-induced TNF, and Type 1 IFN activity resulted in the apoptosis of regulatory T cells and aggravation of steatohepatitis induced by the MCD diet [119]. The researchers concluded that TLR-7 signaling can induce TNF production in KC and type I IFN production in dendritic cells, leading to hepatocyte death and the progression of NASH [119]. Mice fed an MCD diet display enhanced TLR4 expression and markers of inflammation. This TLR4 expression can be blunted by the depletion of KC using clodronate. In contrast, TLR4 KO mice display less liver injury and lipid accumulation [120].

Therefore, in NASH, apoptosis seems to be the predominant mode of cell death, as there is evidence for the involvement of the intrinsic pathway of apoptosis (via lipotoxicity and organelle stress), as well as the extrinsic pathway of apoptosis (via cell surface receptors).

### 3.2. MPT-Mediated Necrosis and Necroptosis in NASH/NAFLD

The contribution of necroptosis to the pathogenesis of NASH is the subject of much interest [27,29]. This has been amplified by recent studies showing a worsening of ballooning and fibrosis in clinical trials of NASH using pan-caspase inhibitors, raising the possibility of a switch to a necroptotic form of cell death [112]. However, the possible occurrence of necroptotic cell death under lipotoxic conditions and metabolic stress in hepatocytes is yet to be rectified with the presence of intact caspase machinery in patients.

The role of MPT-derived necrosis in NASH/NAFLD still remains unclear. It has been shown that CypD disrupts calcium balance and induces an exaggerated mitochondrial permeability transition pore (MPTP) opening [16]. CypD-KO mice demonstrate reduced mitochondrial stress, TG content, and steatosis induced by HFD when compared to WT. Hepatic adenoviral overexpression of CypD also promotes steatosis, meanwhile unaltering the serum levels of fatty acids and TGs [16]. To further interpret the role of CypD in NASH, CRV431, a pan-cyclophilin inhibitor, was given to C57BL/6 mice on an HFD [121]. CRV431 caused a significant difference in NASH scores, including steatosis, inflammation, and ballooning after 3–14 weeks of treatment. Fibrosis was also reduced in mice that received longer treatment (20–30 weeks) [121].

Much of the early work studying the role of necroptosis in fatty liver focused on the use of the non-specific RIPK1 inhibitor Nec-1 and globally deficient RIPK3 mice. Furthermore, the increased expression of the RIPK3 protein in whole liver by Western blotting and immunostaining was taken as a surrogate for the activation of necroptosis [27]. Increased RIPK3 expression by immunostaining has been detected in human liver biopsy specimens of patients with NASH [27,122,123]. This led to the examination of transgenic animals, mainly RIPK3 KO and subsequently MLKL KO mice, using dietary models of NAFLD/NASH with some mixed and contradictory results. Roychowdhury and colleagues used an HFD for 12 weeks and detected increased RIPK3 and pMLKL staining in WT mouse livers. The study further compared global RIPK3 KO mice to WT controls and surprisingly found an exacerbation of steatosis, inflammation, liver injury, and hepatocyte apoptosis (positive TUNEL staining) in the RIPK3 KO mice [124]. Interestingly, the RIPK3 KO mice were glucose intolerant, even on a chow diet [124]. Gautheron and colleagues have also observed an induction of RIPK3 following MCD diet feeding, which is further amplified with the hepatocyte deletion of CASP8 [123]. The combined hepatocyte deletion of CASP8 together with global RIPK3 KO led to increased hepatic TGs but reduced inflammatory cell infiltration and decreased fibrosis [123]. RIPK3 has also been shown to be increased in the white adipose tissue (WAT) of obese mice fed a choline-deficient HFD (CD-HFD) [125]. Interestingly, the deletion of RIPK3 in these mice did not prevent cell death and led to increased CASP8-mediated apoptosis, suggesting that the increased RIPK3 was not an indication of necroptosis but likely due to metabolic signaling. Similar to the RIPK3 mice fed a chow diet, in the study conducted by Roychowdhury et al., these mice also suffered from impaired glucose tolerance [125]. Therefore, RIPK3’s effect on glucose homeostasis seems to be another necroptosis-independent role for the protein and should be further studied. Afonso and colleagues have also suggested that RIPK3 expression increases in many forms of liver diseases, including hepatitis B, hepatitis C, alcoholic steatohepatitis, and NASH [126]. Using a CD-HFD and an MCD diet, Afonso et al. concluded that RIPK3 deficiency attenuates liver injury, steatosis, inflammation, oxidative stress, and fibrosis [126].

Xu and colleagues examined the impact of HFD on MLKL KO mice and reported an induction in MLKL, RIPK1, as well as increased pMLKL in the livers of mice fed an HFD [127]. They also detected an increase in RIPK1, RIPK3, and pMLKL in the livers of OB/OB mice [127]. Furthermore, the inhibition of RIPK1, RIPK3, by pharmacological inhibitors or MLKL KO enhanced insulin signaling in vitro and in vivo, which was proposed to occur through MLKL binding the phosphatidylinositol phosphates (PIP) at the plasma membrane [127]. A lack of MLKL or treatment with Nec-1 had no effect on inflammation, while MLKL deficiency had no effect on cell death in vivo [127]. Whether the endosomal sorting complexes required for transport (ESCRT)-III machinery, which is known to interact with MLKL for exosome formation to dampen pMLKL and promote cell survival, contributes to the development of insulin resistance remains to be determined. This study highlights a non-necroptotic function for MLKL that had not been previously known.

In vitro MLKL KO using Crispr-Cas9 gene editing has been shown to lead to increased β-oxidation and mitochondrial mass (through PGC1α) of steatotic hepatocytes [128]. In vivo, treatment with the RIPK1 inhibitor, RIPA-56, prevented inflammation and decreased fibrosis in mice fed an HFD [128]. Furthermore, increased circulating levels of RIPK1 and MLKL were reported in sera of patients with NAFLD and somewhat correlated with ALT levels and histologic activity [128].

Wu et al. have investigated the role of RIPK3 and MLKL using a high-fat, high-fructose, high-cholesterol (FFC) diet [129]. FFC feeding for 12 weeks induced both RIPK3 and MLKL; however, only MLKL KO mice were protected from FFC diet-induced steatohepatitis [129]. Palmitic acid treatment in vitro resulted in MLKL membrane staining in the AML12 cell line, suggesting MLKL membrane translocation [129]. No pMLKL by western blot (WB) was reported. FFC feeding increased the markers of autophagy P62 and LC3-II as well as the markers of ER stress CHOP and p-eIF2α. These markers were abrogated by MLKL KO, suggesting a role for MLKL in regulating autophagic flux [129]. Interestingly, the authors report an RIPK3 independent pathway for MLKL activation and translocation to the cell membrane. The inhibition of autophagy by leupeptin in vivo or chloroquine in vitro triggered MLKL translocation to the plasma membrane, suggesting that MLKL is closely involved with autophagy [129]. The relationship between autophagy and MLKL in NASH and how it is related to MLKL’s necroptotic role remains to be explored further.

It is well known that necroptosis requires the inhibition of caspase activity. Therefore, the occurrence of necroptosis in human NASH/NAFLD in the presence of intact caspases still remains an intriguing event to be elucidated. Nevertheless, there seems to be an accumulating body of evidence pointing to a role for RIPK1, RIPK3, and MLKL in fatty liver disease. RIPK1 and RIPK3 have known non-necroptotic functions [130]. The lack of littermate controls, the use of polyclonal antibodies, and various dietary models may have contributed to some of conflicting reports in the NASH models [27]. Additionally, in most of these studies, RIPK3 is globally knocked out and is absent in the liver immune cell, KC and endothelial cell compartment, as well as hepatocytes. RIPK3 deletion and catalytically inactive RIPK1 have been shown to provide greater benefit than MLKL deletion in various animal models of injury, pointing to extra-necroptotic roles for the RIP kinases [29,131]. Non-necroptotic functions of MLKL have been described in vesicle trafficking, the regulation of inflammasomes, as well as autophagy [31,32,129]. Then, the question arises: Are the RIPK1–RIPK3–MLKL axis proteins affecting NASH through the mediation of necroptosis and promotion of an inflammatory cell death phenotype leading to fibrosis, or do they have alternative metabolic, inflammatory, autophagic and perhaps non-cell death-related activities that drive NASH? Furthermore, the role of these proteins in the non-parenchymal liver cell compartment remains to be explored. In order to get some clear answers and truly study the contribution of necroptosis, the impact of conditional knockout of MLKL in adult hepatocyte needs be examined in an HFD that induced insulin resistance.

### 3.3. Pyroptosis in NASH/NAFLD

Pyroptosis is an inflammatory cell death that is the result of CASP1-mediated inflammasome activation. Inflammasomes are comprised of multiple parts including a sensor, an adaptor protein, and a pro-caspase in zymogen form [89]. When activated, the inflammasome complex regulates the activation of CASP1 by cleavage, which results in the induction and cleavage of inflammatory cytokines IL-1β, IL-18, as well as the effector of pyroptosis, GSDMD [89]. Inflammation is a key feature of NASH, and the NLRP3 and AIM2 inflammasomes have been shown to be activated in liver immune cells under multiple experimental conditions [132]. In NASH, inflammasome activation is thought to be triggered by lipotoxicity, organelle stress, and hepatocyte cell death, resulting in the release of DAMPs, which in turn activate KC and stellate cells [133]. Additionally, the liver’s constant exposure to low level LPS, a PAMP, through the portal venous system may drive inflammasome activation in NASH and other liver diseases. IL-1β’s inflammatory activity is amplified due to its synergistic action with TLR signaling, leading to paracrine and autocrine effects that increase pro-IL1β as well as other cytokines and chemokines, namely TNF and CCL2. This inflammatory milieu ultimately results in the activation of hepatic stellate cells, which promote a fibrogenic phenotype and liver fibrosis [132,133]. Inflammasome-mediated dysbiosis was shown to regulate the progression of NAFLD and obesity in a mouse model using MCD feeding [134]. CASP1 KO, ASC KO, or NLRP3 KO mice were fed MCD for 24 days, and all displayed increased transaminases, worsening inflammation, and steatosis compared to WT controls [134]. The exacerbation of liver injury was also evident in IL-18 KO mice but not in IL-1 receptor KO mice [134]. When NLRP3 deficiency was limited to the immune system (bone marrow chimera), the severity of NASH was not significantly worse than WT animals fed an MCD diet for 24 days [134]. Likewise, constitutively expressed NLRP3 in hepatocytes or CD11c myeloid cells did not result in any significant differences in MCD-induced NASH compared to WT littermates [134]. Interestingly, the cohousing of CASP1 KO, ASC KO, NLRP3 KO, and IL-18 KO mice with WT animals before induction of NASH with an MCD diet resulted in significant exacerbation of NASH in the WT cage mates, suggesting that the development of NASH could be transmissible via gut microbiota [134]. TLR4 KO and TLR9 KO mice cohoused with the inflammasome-deficient mice did not exhibit any increased disease severity, suggesting that bacterial products from the portal system are triggering the TL4/9 receptors driving liver inflammation and the progression of NASH [134]. Microbiome dysbiosis was the key contributor to the findings, and a significant expansion of Porphyromonas species was observed following MCD and HFDs, which was abolished with antibiotic treatment [134]. The effect of inflammasome deficiency, the observed intestinal dysbiosis, and their relationship with hepatocyte cell death remain unclear. The effect of KO on these inflammasome constituents and the resultant increase in PAMPs in the portal circulation on the mode of hepatocyte cell death and in particular on GSDMD and pyroptosis should be explored. 

Mridha et al. used an NLRP3 inhibitor, MCC950, in two murine steatohepatitis models, the HFD-fed *foz/foz* mice (which are appetite defective and overeat) and the MCD diet [135]. MCC950 prevented the rise in AST and ALT in *foz/foz* mice and in WT mice and resulted in decreased NFκB activation, leading to less liver inflammation and overall improvement in the NAFLD activity score (NAS) score, without any effect on steatosis as observed on histology [135]. MCD-fed mice had higher active CASP1 and IL-1β, which were reduced with NLRP3 inhibition, but there was no effect on IL-18. NLRP3 inhibition delayed hepatic fibrosis with lowered fibrotic markers and macrophage and neutrophil infiltration in both NASH murine models [135]. While cell death and pyroptosis were not directly studied here, a strong trend toward decreased ballooned hepatocytes in the mice treated with the inhibitor suggests a possible decrease in cell death [135]. Xu et al. investigated the effects of GSDMD, the principal executioner of pyroptosis downstream of CASP1, 4, 5, and 11, in NAFLD patients and multiple animal models of NASH [47]. The levels of cleaved GSDMD-N fragment as determined by Western blotting were upregulated in the liver biopsy specimens of patients with NASH and correlated with the activity score and fibrosis [47]. GSDMD expression was increased in the livers of db/db mice fed an MCD diet [47]. GSDMD KO mice fed an HFD for 11 weeks had significantly decreased steatohepatitis, lower ALT and TG levels, and ameliorated liver fibrosis [47]. In addition to improved liver histology, the KO of GSDMD resulted in an improved hepatic cytokine profile with reduced levels of TNF, MCP-1, and IL-1β [47]. Further research using GSDMD KO models with other dietary models is needed to study the role of pyroptosis in NAFLD/NASH [47,89,136].

### 3.4. Ferroptosis in NASH/NAFLD

The contribution of ferroptotic cell death to NAFLD/NASH is not clear. Lipid peroxidation, ROS accumulation, and increased liver iron stores (both in the parenchymal and non-parenchymal compartment) have been noted in patients with NASH [137]. Furthermore, antioxidants such as vitamin E and iron reduction by phlebotomy have been shown to improve liver chemistries, which is a surrogate for hepatitis, in patients with NAFLD/NASH [138,139]. In a recent study, Qi et al. observed an induction of GPX4 protein after 24 weeks of Western diet or 10 days of MCD feeding in mice [140]. Treatment of WT mice fed an MCD diet for 10 days with RSL-3 (a ferroptosis inducer) resulted in decreased hepatic GPX4 but aggravated hepatic steatosis with increased ALT levels and histologic inflammation [140]. The induction of ferroptosis correlated with increased pro-inflammatory cytokines TNF, IL-6, and IL-1β. This was deemed to be through apoptosis-inducing factor (AIF)-mediated cell death activation due to increased lipid peroxidation by lipoxygenase [140]. Increased TG accumulation was observed by Oil red o staining and increased hepatocyte death was confirmed with TUNEL staining [140]. Treating mice with sodium selenite (SS), a GPX4 activator, reversed this effect by promoting cell survival, reducing the appearance of NASH on histology, improving AST and ALT, and inflammatory cytokine levels. The authors concluded that the modulation of ferroptosis and GPX4 could be an innovative therapeutic strategy for NASH [140].

Another study assessed the contribution of ferroptosis versus necroptosis in NASH using a choline-deficient, ethionine-supplemented diet (CDE). Within 1–2 days of CDE feeding, a drastic accumulation of lipids, infiltration of inflammatory cells, and elevated transaminases were detected [141]. The authors explored the cell death subroutine by in vivo propidium iodide (PI) staining, as well as immunohistochemical staining with cleaved CASP3 [141]. The mode of cell death was deemed to be necrosis, but not necroptosis, as MLKL KO mice fed with the CDE diet displayed no protection from cell death [141]. Administration of Trolox, a vitamin E analogue reported to be a ferroptosis inhibitor, decreased necrotic cell death and inflammatory cytokines. The authors concluded that ferroptosis is possibly contributing to the observed necrotic phenotype of cell death. The type of diet used, as well as the short duration of therapy and the time point studied, are not representative of human disease pathology. Therefore, conclusions regarding the cell death mode in NASH, which is a chronic disease in patients with metabolic syndrome and obesity, using this model are debatable. Furthermore, the specificity of Trolox, which is an antioxidant, to ferroptosis is questionable. It is possible that the model induced ROS-mediated MPT-driven cell death, which was inhibited by the addition of the antioxidant and vitamin E analogue. Other ferroptosis inhibitors have been studied in the MCD diet and shown to improve lipid peroxidation, inflammation, liver injury, and fibrosis [142]. Furthermore, mitochondrial morphological changes with an MCD diet, such as organelle shrinkage and increased membrane density, have been observed with MCD diet feeding, which were reversed with Ferrostatin treatment [142]. Ferrostatin also decreased hepatic steatosis, inflammatory cytokines, and fibrosis [142].

Future studies are required to verify the potential role of ferroptosis and GPX4 signaling in NAFLD and NASH using more physiologically relevant animal models that display insulin resistance. However, it is clear that ROS accumulation and lipotoxicity are central features of NASH, and ferroptosis inhibitors are radical scavengers and antioxidants. Therefore, manipulating pathways that affect the GSH and ROS balance in the liver could contribute to NASH disease activity and progression. Whether the term ferroptosis applies or this is classical cell death due to oxidative stress/lipotoxicity is a matter of semantics.

## 4. Cell Death in Acetaminophen Toxicity

Drug-Induced Liver Injury (DILI) can be due to direct toxicity from drugs or due to an idiosyncratic reaction that occurs in a small subset of susceptible individuals [143]. The molecular mechanisms for idiosyncratic DILI are thought to be the result of an interplay between a drug or a small molecule’s chemical structure activating the adaptive immune system in individuals with HLA polymorphisms and a secondary predisposition such as lack of adaptation [143]. This has been reviewed elsewhere in detail [144].

Here, we will focus on cell death from APAP hepatotoxicity, which is the most prevalent etiology of direct hepatotoxicity and the most common cause of acute liver failure in the U.S. [145]. APAP-induced hepatocyte death is a form of regulated necrosis that starts in the centrilobular region or zone III of the liver. Mouse models of APAP toxicity closely mimic human pathology and recapitulate this injury. These valuable tools have been used to shed light on the molecular signaling cascade leading to APAP cell death. In recent years, various forms and subroutines of cell death have been studied using this model with controversial results [146]. Currently, APAP is firmly believed by experts in the field to be a form of MPT-mediated, regulated hepatocyte necrosis [9,146,147].

### 4.1. MPT-Mediated Regulated Necrosis in APAP

APAP is converted in the cytosol into its toxic metabolite NAPQI (N-acetyl-*p*-benzoquinone imine) via direct oxidation by the CYP450 enzymes Cyp2E1, Cyp1A2, Cyp3A4, and Cyp2D6 [148]. NAPQI is a highly reactive metabolite that is efficiently detoxified by GSH forming, APAP–GSH conjugates [149]. However, when GSH is depleted and cysteine supply is limited, the highly reactive NAPQI metabolite is free to attack protein thiols and cysteine residues throughout the cell, forming NAPQI adducts that lead to ER stress and mitochondrial damage [63,150]. Supplementation with the GSH precursor, cysteine, in the form of N-acetyl cysteine (Mucomyst™), is an effective antidote to APAP poisoning if given within 10 h of overdose in humans and 1.5–2 h in mice [151]. Despite the excellent correlation between NAPQI–protein adduct formation and toxicity, no direct causality between adduct formation and hepatocyte necrosis has been demonstrated [147]. Recently, the removal of APAP adducts though selective autophagy within 24 h has been suggested to dampen toxicity and cell death [152]. The resultant organelle stress from NAPQI–protein adducts leads to the generation of ROS, hydroxyl radicals via an iron-mediated mechanism (Fenton reaction), as well as peroxynitrite formation through the interaction of superoxide with mitochondrial nitric oxide (NO) [153]. Mitochondrial damage plays a key role in the pathogenesis of APAP-induced necrosis [154]. APAP has been shown to impair mitochondrial respiration and affect the electron transport chain (ETC) both in vitro and in isolated hepatocytes from in vivo treated animals [147]. The generation of ROS subsequently exacerbates mitochondrial stress and may induce ER stress, leading to the activation of intracellular signaling pathways, most importantly the MAPK pathway [153]. Inhibition, knockdown or KO of various MAPK proteins, including mixed lineage kinase protein 3 (MLK3), ASK1, mitogen activated protein kinase 4 (MKK4), JNK, as well as the JNK binding partner SH3BP (Sab) are markedly protective against APAP-induced hepatocyte death [155,156,157,158,159,160] (Figure 2). The ultimate target in the MAPK signaling cascade that promotes APAP cell death is the phospho-activation and mitochondrial translocation of JNK [155]. JNK activation is usually transient, and sustained activation of p-JNK due to ongoing organelle and cellular stress, along with the perpetual ROS species generation, leads to cell death. When stress signals exceed a certain threshold (such as toxic doses of APAP), p-JNK interacts with mitochondria by binding to the kinase interacting motif of Sab, a mitochondrial outer membrane protein. This results in the dephosphorylation of activated -Src through the tyrosine phosphatase SH2 phosphatase 1 (SHP1), and ultimately increases mitochondrial ROS generation by dampening the ETC [160]. This increase in ROS and sustained stress leads to MPT (Figure 2). Mitochondrial membrane rupture leads to the release of AIF and endonuclease G (endo G), which translocates to the nucleus, causing DNA fragmentation [161]. AIF-induced nuclear damage is important in APAP-induced cell death, as AIF-deficient mice have less DNA fragmentation and less injury from APAP [162]. The importance of MPT-mediated necrosis in APAP toxicity and cell death has been bolstered by studies demonstrating inhibitors such as cyclosporine A abrogate APAP toxicity in vivo and in cell culture models [163,164]. Additionally, CypD-deficient mice are also protected from APAP hepatotoxicity and cell death [15]. The mitochondrial fission protein dynamin-related protein-1 (DRP1) translocates to the mitochondria post-APAP, and DRP1 inhibition by MDIVI has also been shown to dampen hepatocyte death from APAP [30,165]. The exact role of DRP1 is unclear, but it can be speculated that DRP1 translocation to mitochondria post-APAP mediates mitochondrial fission. Upstream of the MAPKs and JNK, other kinases and signaling molecules have been implicated in hepatocyte death from APAP. The knockdown or inhibition of glycogen synthase kinase 3 beta (GSK3β) and RIPK1 have been effective in dampening cell death from APAP [30,166,167] (Figure 2). RIPK1 participates in APAP toxicity upstream of JNK and is independent of its role in necroptosis [27,30]. The role of RIPK3 is more controversial and will be discussed in detail below.

### 4.2. Apoptosis in APAP

APAP-mediated hepatocyte death results in a lytic form of cell death typical of necrosis. In fact, hepatocytes that die of APAP toxicity tend to not display any of the typical characteristics of apoptotic cells such as shrinkage, nuclear condensation, blebbing, and karyorrhexis. Despite this, many investigators have investigated whether APAP cell death could perhaps be apoptotic, at least in the initial phases of injury, and then transition into a secondary necrosis in the later phases. Recently, several studies have claimed that apoptosis plays a role in APAP hepatotoxicity by using TUNEL staining, immunoblots of cleaved caspases, or using caspase inhibitors soluble in DMSO, which is a solvent that itself is a P450 inhibitor affecting APAP metabolism [168,169,170,171,172,173,174].

However, a few key facts make apoptosis unlikely in both animal models of APAP and in vitro [175]. APAP overdose causes mitochondrial damage and profound ATP depletion, preventing caspase activation, which is strongly ATP-dependent [176]. Secondly, caspase inhibitors (when used with appropriate controls) do not protect from APAP-induced cell death [171,174].

Much of the confusion regarding apoptosis in APAP seems to stem from positive TUNEL staining observed on the livers of APAP-treated mice [174]. However, as discussed above, due to the release of AIF and endo G following MPT, some DNA fragmentation from APAP is to be expected. Additionally, the pattern of TUNEL staining from APAP is cytosolic and not the classic nuclear positive TUNEL stain seen in apoptotic hepatocyte death such as in FAS/FASL or TNF-mediated apoptosis [177]. Nuclear laddering in APAP is not accompanied by CASP3 cleavage and is not prevented by caspase inhibitors [177].

Another source of controversy has been from the Bcl2 family. Enhanced APAP toxicity was reported in mice overexpressing Bcl2, which is a traditionally anti-apoptotic protein [178]. Additionally, both cleaved tBid and Bax, Bcl2 proteins that traditionally participate in apoptosis, have been demonstrated to translocate to the mitochondria following APAP [170]. This translocation is dependent on JNK activation and is thought to mediate the release of mitochondrial proteins AIF and endo G [155,161]. However, there is no evidence of MOMP or apoptosis from tBid and BAX translocation to mitochondria during APAP. The mechanism of Bid cleavage in the absence of caspase activation remains unclear, although calpains have been proposed as the responsible proteases [174].

P53 upregulated modulator of apoptosis (PUMA), another BL2 family member, has been shown to participate in APAP-induced hepatocyte death—however, not as a mediator of apoptosis [167]. Lack of PUMA protected hepatocyte death and pre-treatment with RIPK1 or JNK inhibitors, similarly to JNK and RIPK1 small interfering RNA (siRNA) knockdown, abrogated PUMA upregulation. Importantly, treatment with a PUMA inhibitor post-APAP administration mitigated APAP necrosis and cell death. However, the mechanism through which PUMA exerts its effects on hepatocyte necrosis downstream of JNK remains unclear [167].

### 4.3. Necroptosis in APAP

The contribution of necroptosis in APAP has also been explored [27]. Initially, many of these studies were carried out with Nec-1, a RIPK1 kinase inhibitor, which had off target effects and was discovered to be identical to the indoleamine 2,3-dioxygenase (IDO) inhibitor, methyl-thiohydantoin-tryptophan (MTH-Trp), a strong immunomodulator [23,179]. In addition to the off-target effects of Nec-1, the compound is solubilized in dimethylsulfoxide (DMSO), a Cyp450 inhibitor, which makes conducting APAP studies problematic as drug metabolism, and the amount of toxic metabolite generation can confound results [180]. Therefore, mice with genetic deletions such as global RIPK3 knockout mice were developed and used instead. RIPK3′s role in APAP-induced hepatocyte death was also the source of much controversy [27,165]. Conflicting results were generated mainly due to differences in WT controls used, as most investigators did not cross the RIPK3 KO mice, which were viable and had no phenotype with WT mice to use littermate controls [27]. RIPK3 KO mice were reported to be protected against APAP toxicity by several investigators [165,181]. However, our group was unable to detect a difference in hepatocyte injury and cell death in RIPK3 KO mice [30]. Furthermore, we did not see a difference in MLKL KO mice compared to strain-matched WT control mice either, and this lack of protection with MLKL KO was confirmed by others [30,36]. In contrast to RIPK3 and MLKL, RIPK1 protein knockdown via antisense and siRNA has been shown to protect against APAP independently of necroptosis and upstream of JNK signaling [30,167]. Overall, as previously discussed, necrosis in APAP is generally accepted as being MPT-mediated and not necroptosis.

### 4.4. Autophagy in APAP

Autophagic cell death does not seem to play a role in APAP. In fact, the activation of autophagy transcriptionally or pharmaceutically is protective against APAP-induced cell death. On the other hand, preventing autophagy has the opposite effect, as it exacerbates APAP toxicity [182,183,184,185]. Mitochondrial dysfunction, mitochondrial protein adducts, ROS production, and ATP depletion are all implicated in APAP-related necrosis and lead to the activation of autophagy [152,182]. The mitochondrial protein adducts generated as a result of NAPQI activation during APAP toxicity are the main catalysts of mitophagy and clearance of damaged mitochondria. The importance of these mitochondrial adducts was demonstrated with studies using AMAP (acetyl–metal–aminophenol), an isomer of APAP, which when given at toxic doses forms protein adducts exclusively in the cytosol and not in the mitochondria and, unlike APAP, AMAP does not cause liver injury [186,187]. The induction of mitophagy and autophagy is thought to limit ROS generation by damaged mitochondria, thereby limiting the expansion of the necrotic area and promoting mitochondrial biogenesis and liver regeneration [188].

Using LC3 transgenic mice and APAP administration, an increase in autophagosomes has been observed surrounding necrotic areas [152,188]. Ni and colleagues examined these structures using electron microscopy and defined a unique zonated pattern following APAP necrosis [188]. The researchers show that the autophagy zone surrounds a zone with spheroidal mitochondria adjacent to the necrotic debris and hypothesize that these autophagosomes form a barrier to restrict necrosis expansion [188]. APAP increases autophagic flux in primary human hepatocytes, and the inhibition of autophagy increases the highly reactive APAP–protein adducts, indicating a possible translational relevance in these studies [152]. Therefore, the authors suggest that the induction of autophagy post-APAP overdose at the early stages to clear out reactive adducts may have clinical benefits. How to induce limited autophagy safely in the liver remains to be determined.

### 4.5. Pyroptosis in APAP

Inflammation is a key late feature of APAP necrosis. This occurs due to the release of DAMPs such as mitochondrial DNA, high mobility group box 1 (HMGB1), and nuclear fragments following lytic cell death. Pyroptosis in APAP-induced liver injury has not been studied broadly; however, the role of the inflammasome has been investigated [189]. Imaeda et al. demonstrated that *CASP1 KO*, *ASC KO*, and *NLRP3 KO* mice were significantly less sensitive to APAP-induced injury based on histopathology and ALT levels compared to WT mice. However, others have failed to demonstrate any protection or decrease in hepatic inflammatory cell recruitment with anti-IL-1β antibodies or KO of IL-1β, CASP1, or NALP3 [190,191,192]. Zhang et al. propose that IL-1α (not IL-1β) generated through the activation of TLR4 signaling in macrophages is the key mediator of neutrophil and monocyte recruitment to the liver, possibly aggravating injury from APAP [192]. Therefore, the NLRP3 inflammasome does not seem to contribute to APAP DILI. The inflammasome and pyroptosis bring the contribution of innate immunity and the role of sterile inflammation in APAP to the forefront of cell death pathogenesis. Sterile inflammation serves to clean necrotic debris and dead hepatocytes and promote liver regeneration and healing. However, it is important to note that evidence for a second wave of inflammatory cell-mediated liver injury is limited. The contribution of the inflammatory response to the pathogenesis of APAP is controversial and beyond the scope of this review and has been reviewed in detail elsewhere [193].

Yang and colleagues have examined the effect of APAP on GSMD KO mice compared to non-littermate WT control mice and have reported increased injury in mice lacking GSDMD [194]. As GSDMD KO did not protect from cell death, Yang et al. attribute the resultant increase in injury from the prevention of pyroptosis to CASP8-mediated apoptosis and necroptosis [194]. However, as discussed above, neither caspase KO and inhibition nor MLKL KO have any effect on APAP-induced cell death [30,171]. One explanation could be that the authors used the incorrect substrain of WT control mice. The substrains are not clearly reported in the study; however, the donated strain of GSDMD KO mice from Jackson and the laboratory of Professor Feng Shao are on an C57BL6n substrain [195], while the WT mice from Jackson are on a C57BL6j background. Not using littermates and a lack of backcrossing and substrain matching could have possibly contributed to the results seen by Yang and colleagues [196]. Overall, pyroptosis inhibition and GSDMD KO do not seem to prevent APAP-induced liver cell death.

### 4.6. Ferroptosis in APAP

GSH depletion, iron accumulation, and lipid peroxidation have long been identified as the foundation mechanisms of APAP-induced liver injury [197]. However, their participation leading to ferroptotic APAP-induced hepatocyte death is unknown.

Multiple investigators have examined the in vitro effect of the specific ferroptosis inhibitor, ferrostatin 1 (Fer-1) in APAP-induced cell death [198,199]. Not surprisingly, similar to other antioxidants, Fer-1, a free radical scavenger, has been shown to protect hepatocytes from APAP [199]. Schnellmann et al. demonstrated that the chelation of intracellular iron by Deferoxamine (DFO) alleviates APAP-induced liver injury in a dose-dependent manner [200]. Fer-1 has been shown to prevent cell death from APAP both in vitro and in vivo [201]. While there was no impact on CYP2E1 expression with Fer-1, pretreatment with the inhibitor resulted in increased GSH levels 3 h post APAP in mice [201]. The authors saw no cleaved CASP3 or increased RIPK3 expression with Fer-1 treatment. DFO also protected from APAP by maintaining GSH levels post APAP administration; lipid peroxides from arachidonic acid were reported to be the primary cause of injury and cell death [201]. Preventing hepatic GSH depletion is an effective way of protecting against APAP, and whether doing so by preventing ferroptosis is really a matter of nomenclature. Given the controversy surrounding the importance of lipid peroxidation in APAP cell death, the role of lysosomal iron, and the lack of conclusive evidence with Fer-1, future studies using GPX4 KO mice are necessary to determine if ferroptosis is in fact critical to cell death in the APAP model [146,202].

## 5. Cell Death in Autoimmune Hepatitis (AIH)

### 5.1. Apoptosis in AIH

Immune-mediated liver diseases are the outcome of adaptive immune-mediated injury to hepatocytes. Autoimmune hepatitis is a disease most often seen in women, and it is characterized by elevated IgG, seropositivity for various autoantibodies, and chronic hepatitis and piecemeal necrosis with infiltration of plasma cells and lymphocytes on liver biopsy [203].

In liver biopsy specimens of patients with AIH, dead hepatocytes are seen as acidophilic, condensed, and shrunk cells referred to as “councilman bodies” [203]. Due to the apoptotic appearance of councilman bodies, apoptosis has long been considered the mode of cell death in AIH [203]. While no perfect animal model exists to mimic the adaptive immune response seen in human AIH, the murine Concanavalin A (ConA) and *α*-galactosylceramide (*α*-GalCer) have been used as models for immune-mediated liver disease [204,205].

It is important to consider that most of the data discussed here has been observed in these models and does not necessarily translate to cell death in human AIH. Unlike APAP, which is not reliant on external DR signaling, ConA hepatitis, which involves the indiscriminate activation of T cells (through a TCR-independent fashion), as well as natural killer T (NKT) cells, is mediated through cytokines. IFN-γ, IL-4, IL-6, FAS, and TRAIL have all been reported as mediators of ConA hepatitis and cell death, although the principal cytokine contributing to toxicity is known to be TNF [206].

TNF activation does not always result in apoptosis via the JNK–Bim pathway, as pro-survival activation of NFκB is dominant in hepatocytes. In most liver injury models with TNF, transcriptional or translational inhibition of NFκB is necessary for apoptosis to occur. However, this is not true for the ConA model. In this context, TNF can exacerbate cell death during the massive T cell response by sensitizing both primary hepatocytes in vitro and the whole liver in vivo to FasL-induced apoptosis with the transcriptional induction of Fas via the NFκB pathway [27,207].

Fox et al. assessed FasL and granzyme B levels on liver specimens of patients with AIH compared with healthy livers and observed elevated levels of both FasL and granzyme B in AIH patients by PCR and immunoprecipitation compared to normal livers [208]. Employing specific antibodies against FasL, its expression was detected in infiltrating mononuclear cells surrounding the portal triad, although the expression level in hepatocytes was not reported in this study [208]. Despite the limited sample size, apoptotic bodies were detected by TUNEL staining of human liver sections in patients with AIH and primary biliary cholangitis (PBC) [208]. However, apoptosis has not been thoroughly investigated in AIH, and in spite of the clear role for TNF in the ConA model, the morphology of cell death during fulminant AIH flare does not seem to be apoptotic but rather to resemble necrosis, although it could be argued that the lytic necrosis seen in AIH is secondary to apoptosis [27,203]. In experimental models such as ConA, caspase inhibitors have not been protective against liver injury [209]. The necrotic appearance of cell death combined with the importance of DR activation and TNF has directed researchers to assess whether necroptosis is the mode of cell death in ConA and *α*-GalCer induced liver injury [29].

### 5.2. Necroptosis in AIH

Several studies have observed that RIPK1 inhibition by Nec1 protects against ConA liver damage [210,211,212]. Gunther et al. detected increased hepatic MLKL mRNA and protein expression in liver biopsy samples of patients with AIH [36]. Moreover, pMLKL membrane staining in hepatocytes of AIH patients was also detected [36]. Subsequently, the researchers used the ConA model of T cell-mediated hepatitis and observed the same induction and cell membrane translocation of MLKL. MLKL KO but not RIPK3 KO mice were protected from ConA liver damage [36]. Unlike RIPK3 KO, the specific RIPK1 inhibitor, Nec-1s, was protective in ConA-treated mice, and this protection was upstream of MLKL activation. Therefore, an alternative unknown pathway for MLKL activation was hypothesized [213]. Gunther et al. further provided evidence that ConA activates T cells and NKT cells, resulting in increased cytokines such as TNF and IFNγ. The IFNγ–STAT1 pathway was necessary for the induction of MLKL and TNF, as well as for MLKL activation and translocation to the plasma membrane [36,213].

To further understand the role of hepatocyte MLKL, Hamon et al. studied liver-specific conditional MLKL knockout mice (MLKL LPC-KO) and could not find a difference between ConA-treated MLKL LPC-KO and littermate controls [214]. Hamon et al. proposed that these contradictory findings may be due to the effects of MLKL KO on other liver cell populations, such as KC and liver sinusoidal endothelial cells in the MLKL global KO studies conducted by Gunther and colleagues. Both studies were unable to detect apoptosis in MLKL KO or WT mice treated with ConA [36,214]. Based on the work of Hamon et al., it seems that necroptosis is dispensable for ConA-induced cell death. Using RIPK1 kinase-dead knock-in mice (Ripk1-K45A), the RIPK1 inhibitor, Nec-1s (which is a potent and much more specific inhibitor of RIPK1 than Nec-1), and RIPK1 liver parenchymal cell-specific KO (RIPK1-LPCKO), Filliol and colleagues have also explored the effect of Con-A and found that RIPK1-deficient hepatocytes died of TNF and caspase-dependent apoptosis following low dose Con-A injection [215,216]. Kang and colleagues used αGalCer treatment to activate NKT cells and observed less cytokine activation in leukocytes isolated from RIPK3 KO mice. RIPK3 deficiency was shown to have a modest protective effect on injury, not through the prevention of necroptosis or apoptosis but by limiting cytokine production and NK cell activation [217]. RIPK1 levels were induced with αGalCer treatment; however, the administration of Nec-1s did not reduce the expression of IFNγ or TNF, indicating that RIPK1 does not play a role in RIPK3-dependent activation of cytokine production in NKT cells [217]. In another study, Nec-1 and a RIPK1 kinase-dead knock-in (RIPK1-D138N) mice were used to explore the effect of necroptosis in the αGalCer liver injury model [205]. While no effect was seen with kinase dead RIPK1, knockdown of RIPK1 notably intensified liver injury and provoked lethality, unveiling a crucial kinase-independent platform function of RIPK1 [205]. Pretreatment either with a pan-caspase inhibitor (zVAD.fmk) or with anti-TNF antibodies prevented liver injury and lethality [205]. Moreover, RIPK3 KO and MLKL KO mice had similar injury to WT mice, ruling out necroptosis as the cell mode of death [205]. Therefore, there seems to be a kinase-independent platform function of RIPK1 that is protective against TNF-mediated apoptosis in the αGalCer model [205].

The inconsistencies in many of these reports could be reduced by employing standard dosing regimens, set time points, and proper controls, including littermate matching [218]. Additionally, it should be kept in mind that the ConA and *α*-GalCer models are not true representative models of human AIH, and the cell death mode seen in these experimental models may not be seen in patients with AIH [29].

### 5.3. Pyroptosis in AIH

Pyroptosis has been investigated in ConA and *α*-GalCer animal models, which show significant elevation in NLRP3, cleaved CASP1, and IL-1β [219,220]. NLRP3 KO and CASP1 KO mice were protected against ConA hepatitis based on improved histology and reduced transaminases [219]. CASP1 KO mice demonstrated reduced expression of NLRP3, cleaved CASP1, and IL-1β, pointing out that CASP1 was needed for IL-1β production and pyroptosis in this model [219]. Pretreatment with recombinant human IL-1 receptor antagonist (rhIL-1Ra), which blocks IL-1 activity post-ConA liver injury, greatly suppressed hepatitis through reduced TNF and IL-17 secretion and decreased inflammatory cell infiltration [219]. Furthermore, as ROS scavengers such as N-acetylcysteine reduced liver inflammation, it has been suggested that ROS itself is a potential initiator of inflammasomes [219]. However, it is unclear whether pyroptosis is mediating hepatocyte cell death or the death of other cell population(s) in the liver. Lan et al. showed that NKT cells express co-stimulatory TNF superfamily receptor OX40 and high levels of CASP1 [221]. Activation of CASP1 in NKT cells results in processing of pro–IL-1β to mature IL-1β, as well as GSDMD cleavage to yield GSDMD-N, which leads to pyroptotic cell death. Low dose ConA administration (1 mg/kg) into WT C57BL6 mice induced a prominent expression of OX40L in the liver, which was associated with increased serum transaminases, IL-1β and IL-18 [221]. Importantly, treatment with a blocking anti-OX40L monoclonal antibody at the time of ConA injection prevented the hepatitis, the rise in IL-1β and IL-18 expression, and the inflammatory response [221]. Whether ConA and *α*-GalCer liver injury is attenuated in GSDMD KO mice remains to be determined.

### 5.4. Ferroptosis in AIH

Two studies have recently investigated ferroptosis in ConA murine models. One study assessed the upregulation of the gene indoleamine 2,3-dioxygenase 1 (IDO1), which is a heme- containing enzyme associated with ferrous production, in the ConA model of AIH and its function in vitro and in vivo [222]. The studies identified IDO-dependent ferroptosis and the pivotal role of reactive nitrogen species (RNS) in the ConA murine model [222]. IDO1 deficiency activated xCT expression and was accompanied by a reduction in liver lesions and RNS [222]. Another study investigated the role of Caveolin-1 (Cav-1) and RNS in ferroptosis-dependent injury in vitro and in vivo and demonstrated that the downregulation of Cav-1 was followed by more RNS production, while hepatic injury was dampened with Fer-1 administration [223]. Reducing RNS with an ONOO^-^ scavenger or a selective iNOS inhibitor reduced ferroptosis and had a protective effect in this model [223]. No data were given on the role of GPX4 in this model.

## 6. Cell Death in Cholestatic Liver Diseases

### 6.1. Apoptosis in Cholestatic Liver Diseases

Cholestatic liver diseases, such as the classic cholangiopathies PBC and primary sclerosing cholangitis (PSC), drug toxicity, or obstructive processes, result in the impedance of bile excretion termed cholestasis. In cholestatic conditions, the accumulation of toxic bile salts is believed to lead to ligand-independent activation of death receptors and subsequent apoptotic cell death of cholangiocytes [224]. Cholangiocyte death leads to inflammation, liver injury, and fibrosis. Indeed, cholangiocyte apoptosis characteristics such as cytoplasmic shrinkage, nuclear condensation, and classic apoptotic TUNEL positivity have all been observed in liver biopsies of patients with PBC [225]. The pathway involved is thought to be FAS/CD95 mediated, as FAS expression has been detected using immunostaining in the cytoplasm of bile duct cells of PBC patients with increased expression of FAS ligand in surrounding lymphocytes and other immune cells [225,226]. Indeed, bile acids can directly induce apoptosis in cultured hepatocytes, which is a model that can also be mirrored in vivo by bile duct ligation (BDL). Inhibition of apoptosis and improved survival and liver injury is observed in FAS-deficient lpr mice [227]. FAS is not the only DR implicated in cholangiocyte apoptosis. Takeda et al. have shown that Trail/DR5 expression is increased in livers of patients with PBC and PSC, and that DR5 KO mice were resistant to bile duct ligation-induced cholestatic cell death [228]. Moreover, anti-DR5 monoclonal antibody treatment in C57BL6 mice resulted in jaundice, cholangitis, and apoptosis of cholangiocytes [228]. Downstream of the DRs, CASP8-mediated cleavage of Bid results in the generation of tBid and subsequent mitochondrial translocation and MOMP through interaction with Bax/Bak [229]. Ursodeoxycholic acid (UDCA), a hydrophilic and non-toxic bile acid, is the primary pharmacological treatment for cholestatic diseases, and it has been shown to protect against toxic bile acid apoptosis [230]. Using a taurine-conjugate of UDCA (TUDCA), Shoemaker and colleagues demonstrated that in vitro, TUDCA could inhibit apoptosis at any time point independently of CASP8 function by promoting pro-survival signals through p38, ERK, MAPK, and the phosphatidylinositol-3 kinase (PI3K) pathway [230].

Further evidence for an apoptotic model of cell death contributing to cholangiopathy was recently demonstrated by studies on ductular reactive cells. The ductular proliferation or reaction seen with cholangiopathies and bile duct obstruction is a source of inflammation and fibrosis in cholestatic liver disease [231]. Multidrug resistance 2 (MDR2) KO mice crossed with TRAIL KO mice (MDR2/TRAIL DKO) displayed an abundance of ductular reaction, advanced fibrosis, and liver injury [231]. This expansion was attributed to impaired apoptosis, and the BCL2 protein myeloid cell leukemia 1 (MCL1), a regulator of TRAIL apoptosis, was implicated [231]. Treatment of organoids derived from MDR2/TRAIL DKO with an MCL1 inhibitor induced apoptosis in these ductal cells and reduced fibrosis [231]. Cubero et al. reported the overexpression of cleaved CASP3 and CASP8, as well as RIPK3 in liver biopsies of PBC patients, indicating an activation of apoptosis [232]. Using the BDL model in liver-specific CASP8 KO mice, the investigators reported improved AST/ALTs and decreased CASP3 cleavage, pJNK, RIPK1, and RIPK3 levels [232]. KO of CASP8, but not a pan-caspase inhibitor, protected from apoptosis, and no switch to alternative forms of cell death such as necroptosis was observed [232]. In this study, Nec-1 inhibition did not affect liver injury. Interestingly, RIPK3 KO mice with BDL had decreased transaminases, cleaved CASP3, and TUNEL positivity, suggesting a role for RIPK3-mediated apoptosis [232]. Others have shown an attenuation of apoptosis with less CASP3 and 7 positive cells, inflammation, hepatic stellate cell activation, fibrosis, portal hypertension, and even improved survival using pan-caspase inhibitors [233,234]. These findings confirm apoptosis as the likely mode of cell death in BDL mice.

### 6.2. Necroptosis and Necrosis in Cholestatic Liver Disease

Woolbright and colleagues have argued that cell death in cholestatic liver disease is necrotic [235], as neither mouse nor human hepatocytes undergo appreciable apoptosis when exposed to a low concentration of glycochenodeoxycholic acid (GCDCA) and require dramatically higher concentrations [235]. GCDCA caused the cleavage of CASP3 in rat hepatocytes but not primary human hepatocytes, thereby suggesting a species difference. While rat hepatocytes die at concentrations close to 50 μM, human hepatocytes are resistant to GCDCA until much larger concentrations (over 500 μM) [236]. Additionally, the limited detection of caspase-cleaved cytokeratin-18, which is associated with apoptosis in human hepatocytes treated with toxic bile acids, accompanied with the dramatic increase in full-length cytokeratin-18 release associated with necrosis, led the investigators to suggest necrosis as the mode of cell death in human cholestasis [236]. It is true that cholestatic liver disease models can induce areas of necrotic appearing cell death, which is displayed by bile infarcts after BDL or in Mdr2 KO mice [237]. 

Afonso et al. report high levels of RIPK3 and MLKL expression in liver specimens of patients diagnosed with PBC. Using the BDL mouse model, the researchers proceed to show that there is increased pMLKL/MLKL three days post ligation, suggesting that necroptosis may be an early event in this murine model [238]. Interestingly, while RIPK3 KO prevented necroinflammation in the liver in this model, it did not affect apoptosis and subsequent secondary fibrosis. Additionally, RIPK3 KO mice only had less injury 3 days post BDL, as by day 14, the protection was abolished [238]. In a follow-up publication, the authors suggested that miRNA21 mediated necroptosis in the BDL model, as miRNA-21 ablation protected against liver injury and necroptosis likely through cyclin-dependent kinase 2-associated protein 1 (CDK2AP1) [239]. Furthermore, while pMLKL expression was decreased in livers of miR-21 KO mice 3 days post BDL, at 14 days post BDL, pMLKL was similar between WT and miR-21 KO animals [239]. There were a few confusing findings to note: first, pMLKL, CASP3, and CASP8 activity were all increased in the KO mice undergoing sham surgery; second, miRNA-21 KO mice undergoing BDL displayed decreased cleaved CASP3, pJNK, and less TUNEL staining, which is indicative of decreased apoptosis. [239].

While there is circumstantial evidence of necroptosis in the BDL cholestatic model supported by protein expression patterns on liver immunostaining and WBs, since CASP8 signaling is intact, this occurrence remains to be explained. The global RIPK3 KO mouse model used makes it difficult to interpret if hepatocyte, cholangiocyte or non-parenchymal cell RIPK3 ablation results in the favorable outcome post BDL. Furthermore, to make firm conclusions on necroptosis, future studies using conditional adult cell specific knockouts of MLKL would be informative.

### 6.3. Pyroptosis in Cholestatic Liver Disease

The role of pyroptosis and cholestatic liver injury has been studied recently by examining the key pyroptotic mediators, NLRP3 inflammasome and GSDMD. NLRP3 was reported to be expressed in reactive cholangiocytes in patients with PSC, as well as in a murine model of PSC in which mice are treated with 3,5-diethoxycarbonyl-1,4-dihydrocollidine (DDC). In the DDC model, microbial products such as LPS have been shown to potentially have a role in the pathogenesis [240]. NLRP3 expression was increased in PSC patients’ liver biopsies as well as in cholangiocytes of mice treated with DDC [240]. NLRP3 activation leads to increased expression of IL-18, IL-6 (but not IL-1β), and inflammation [240]. Gong et al. reported a dose-dependent induction of the NLRP3 inflammasome in response to chenodeoxycholic acid (CDCA), which is a significant bile acid in cholestatic liver injury [241]. This led to the secretion of macrophage-derived IL-1β by promoting ROS production [241]. CASP1 inhibition in vivo decreased mature IL-1β production and ameliorated fibrosis in the BDL mouse model. The authors concluded that the bile acids can serve as an internal signal, activating the inflammasome to promote pyroptosis [241]. In another study using the OVAbil mouse model of antigen-mediated cholangitis, NLRP3 loss unexpectedly led to a more severe cholangitis phenotype [242]. The OVAbil/NLRP3 KO mice had a severe histopathologic phenotype with loss of bile ducts, larger inflammatory centers, and greater levels of inflammatory cytokine (IL-6 and CXCL10) activation. This was exaggerated when the mice were fed a HFD [242]. NFκB can prime the NLRP3 inflammasome for the activation and propagation of pyroptosis [243].

An Mdr2KO mouse model of PSC was used to demonstrate NLRP3 inflammasome activation within the gut–liver axis [244]. MDR2 KO mice displayed both cleaved CASP8 and CASP3 markers of apoptotic cell death [244]. Additionally, increased NLRP3 activation, ASC adaptor expression, mature IL-1β, and activated CASP1 levels were observed in the MDR2 KO mice, which were deemed to be a result of intestinal dysbiosis [244]. Microbial transplant from MDR2 KO into WT animals resulted in significant liver injury in the previously healthy recipients. Hepatocyte-specific CASP8 deletion did not rescue the phenotype [244]. However, the pan-caspase inhibitor IDN-7314 abrogated liver injury as evidenced by decreased transaminases, less inflammation on histology, decreased serum bile acids, and decreased CASP3 and CASP8 activation in whole liver extracts [244]. Notably, there was no switch to necroptosis or increased cell death with the pan-caspase inhibitor [244]. Treatment with the pan-caspase inhibitor also decreased the protein and mRNA expression of the inflammasome components NLRP3, ASC, IL-1β, and CASP1 [244]. The discrepancy observed between conditional KO of CASP8 in hepatocytes not being protective, combined with the efficacy of pan-caspase inhibition and the downstream effects on inflammasome activation, suggests that the apoptosis of non-parenchymal cells or perhaps intestinal epithelial cells may contribute to the pathogenesis. Strikingly, the pan-caspase inhibitor not only improved liver parameters but also affected the microbiome environment [244]. Whether this change in the microbiome is the result of less injury and inflammation or the driving force behind the liver damage is not clear. The deletion of TNF receptor in MDR2 KO mice (TNFR1 KO/MDR1 KO) aggravates hepatitis and fibrosis [245], and it is accompanied with increased RIPK3 but not pMLKL. RIPK3 expression correlated with the increase in CX3CR1 chemokine, and a significant increase in RIPK3 was seen in CD11b+CXCR1+ monocytes in the livers of TNFR1 KO/MDR1 KO mice, indicating a necroptosis-independent and inflammatory role for RIPK3 [245].

In addition to GSDMD, which is known to be activated by caspases 1,4,5,11, another Gasdermin called DFNA5/Gasdermin E (GSDME) can be specifically cleaved by CASP3 also generating an N-terminal active form that targets the plasma membrane to induce pyroptosis [246]. Therefore, GSDME can link apoptosis and pyroptosis. Xu et al. used a new inhibitor to block CASP3-mediated PARP cleavage and GSDME activation by CASP3, thereby preventing both apoptosis and pyroptosis in BDL mice [246]. Mice treated with the inhibitor displayed less inflammation, lower transaminases, and less cell death [246]. Interestingly, since this inhibitor is rather specific to CASP3 and therefore GSDME, GSDMD should remain able to execute pyroptosis in this model. This would suggest that there is no contribution of GSDMD to cell death in the BDL model and no redundancy between the two gasdermins. Future studies should focus on exploring the effect of GSDMD deletion with or without GSDME to determine which gasdermin is truly the effector of cell death.

## 7. Cell Death in Viral Hepatitis

### 7.1. Apoptosis in Viral Hepatitis

Apoptosis is a mechanism of protection against viral pathogens provoked by host immune defense or the viral proteins themselves. Apoptosis of liver cells is thought to play an important role in the pathogenesis of hepatitis C virus (HCV) and hepatitis B virus (HBV) [247,248].

Histologically, apoptosis with the characteristic councilman bodies, cell shrinkage, and DNA fragmentation as detected by TUNEL staining has long been recognized in the liver sections of patients with viral hepatitis [249]. Numerous death pathway ligands and receptors have been implicated as mediators of apoptotic hepatocyte death in viral hepatitis, including FAS, TRAIL, and TNF [250,251,252]. FAS overexpression in hepatocytes along with increased FASL expression in lymphocytes have been described with HBV and HCV infection, and they have been thought to be a major cause of cell death during active hepatitis [249,253,254,255].

Viral infections such as HBV and HCV have been shown to trigger the ER stress response [63]. This can activate the intrinsic mitochondrial pathway and result in JNK and CHOP-mediated apoptosis [63,256]. Additional evidence for the importance of apoptosis in viral hepatitis was presented by Bantel et al., where elevated levels of cleaved caspases in HCV patients were detected via IHC and WB and significantly correlated with necroinflammatory activity but not with viral load and serum transaminases [257]. To further investigate the role of caspases in viral hepatitis, two studies employed pan-caspase inhibition, resulting in significantly reduced transaminases without affecting the mean viral load [258,259]. Interestingly, T cell apoptosis in HCV infection has been implicated in disease pathogenesis [260]. A strong association between increased T cell apoptosis and immune reactions such as cryoglobulinemia was noted in HCV-infected patients, which was found to be due to decreased NFκB sensitizing activated lymphocytes to cell death [260]. In cell culture models, various HCV and HBV proteins have been shown to influence TNF, BCL2, P53, and NFκB signaling, resulting in cell death, apoptosis, or proliferation-promoting injury [261,262,263]. Many of these studies examining individually overexpressed viral proteins are carried out in vitro using cancer cell lines, and most exhibit substantial variability and inconsistent findings that render drawing firm conclusions difficult. However, in vivo and in human liver biopsy samples, a clear upregulation of DR signaling pathways has been seen; therefore, cytotoxic T cell-mediated cell death via DRs seems to be the mode of cell death in hepatocytes targeted for clearance [263,264,265,266,267]. More recently, studies examining the role of RIPK1 in a viral hepatitis model using murine hepatitis virus type 3 (MHV3) have found that parenchymal cell deficiency of RIPK1, RIPK1LPC-KO, sensitizes mice to apoptosis from MHV3 and PAMPs such as poly I:C. RIPK1 floxed mice also displayed apoptosis as evidenced by cleaved CASP3, although to a lesser extent than mice lacking RIPK1 [268]. The injection of a decoy TNF receptor and depletion of resident liver macrophages was shown to be protective against apoptosis, suggesting that the cell death subroutine is TNF-mediated apoptosis that can be potentiated in the absence of RIPK1 [268].

### 7.2. Necroptosis in Viral Hepatitis

A few studies have investigated whether necroptosis contributes to the pathogenesis of viral hepatitis or acute-on-chronic hepatitis B liver failure (ACHBLF) [269,270]. Chen et al. measured serum RIPK3 levels in ACHBLF patients due to HBV and found significantly elevated levels of RIPK3 and MLKL by IHC in the patient livers compared to healthy controls. Although the source of the elevated levels remains unclear, the levels of RIPK3 detected correlated with survival [269]. Interestingly, while the RIPK3-positive staining on IHC was prominent in the liver sinusoidal endothelial cells and KCs of healthy control livers, it was also detected in hepatocytes appearing to be necrotic or damaged of patients with ACHBLF [269]. Activated MLKL (pMLKL), the effector of necroptosis, was not evaluated in the study, which would have been beneficial in drawing conclusions regarding the involvement of necroptosis. Other studies have also detected an induction of RIPK3 expression on liver biopsies of patients with chronic viral hepatitis, which was not present at baseline [126]. Han et al. measured mRNA levels of RIPK3 in peripheral blood mononuclear cells (PBMCs) isolated from patients with acute on chronic viral hepatitis leading to liver failure and compared them to mRNA levels from patients with chronic hepatitis and healthy controls [270]. Indeed, RIPK1 mRNA levels in PBMCs from patients with acute liver failure were elevated. Not surprisingly, these patients had a very inflammatory cytokine profile with high circulating TNF levels [270]. Circulating levels of total MLKL were also elevated in those with acute on chronic liver failure, and PBMC RIPK3 mRNA levels correlated with TNF, MLKL, and overall survival [270]. This study raises some interesting questions. First, the study investigated acute liver failure, which is a highly inflammatory condition and very different from chronic hepatitis B or C infection. Both RIPK3 and RIPK1 have been shown to have functions independent of necroptosis in inflammation. Second, whether the necroptotic death of circulating PBMCs affects the pathogenesis of hepatitis B and liver cell death is not clear. It is feasible that the necroptotic death of immune cells would release more DAMPs, thus exacerbating the inflammatory milieu in the liver and perhaps promoting more destruction and hepatocyte death. However, the link between the two was not explored, and the mode of cell death in these liver cells remains unclear.

### 7.3. Pyroptosis in Viral Hepatitis

The participation of pyroptosis through NLRP3 inflammasome and CASP1 has been suggested in some small studies. Kofahi et al. explored the effect of pan-caspase, CASP1, and NLRP3 inhibitors on a tissue culture-adapted strain of HCV (JFH1T) in Huh-7.5 cells. Both apoptotic cell death and CASP1-mediated pyroptosis were suggested to occur [271]. Pyroptosis by various cell populations(s) may play a part in HCV liver injury. Monocytes respond to cells with HCV replication by inflammasome-mediated IL-18 production, activation of NK cells, and subsequent IFN responses [272]. Furthermore, viral RNA can induce IL-1β mRNA expression through TLR7. NLRP3 inflammasome stimulates IL-1β generation, leading to amplified inflammation signals associated with HCV disease severity. Indeed, both resident KCs and PBMC-derived macrophages produce IL-1β when exposed to HCV [273]. Similar to the ability of HCV to induce inflammatory cytokines, HBcAg has been shown to influence IL-18 secretion by inducing CASP1 in PBMCs [274]. In contrast, some studies have demonstrated that HBV, and in particular HBeAg, can suppress NLRP3 and IL-1β expression and maturation through inhibition of the NFκB signaling pathway and ROS production [275]. This could be one of the ways in which HBV evades the immune system. The expression of GSDMD-N fragment has been reported to be increased in human liver samples of patients with acute liver failure due to hepatitis B [276]. Further work is needed to understand the contribution of pyroptosis to liver cell death or immune cell death in viral hepatitis.

## 8. Discussion

Cell death is a major driving force for liver inflammation, leading to liver injury, fibrosis, and hepatocellular carcinoma [237]. Over the past few decades, various regulated cell death subroutines with intricate signaling pathways have been discovered [1]. Numerous investigators have studied these cell death pathways by deploying inhibitors and techniques in genetic manipulation in vitro and in mouse models to understand the dominant cell death subroutine in liver disease models (Table 1). Knowing the precise signaling pathways that lead to cell death in each disease will enable scientists to target therapeutics to specific molecules and enzymes alleviating the subsequent injury and inflammation.

Apoptosis was the first described RCD, and therefore, there are more data with this mode of cell death in various liver diseases. Liver cells express caspases and DRs such as FAS, TNF, and TRAIL, all of which have recognized roles in liver injury [60]. Therefore, receptor-mediated extrinsic apoptosis has been implicated especially in liver diseases precipitated by the immune system such as AIH, PBC, and viral hepatitis [60,203]. Furthermore, many liver diseases display shrunken and condensed hepatocytes (councilman bodies), pointing to the occurrence of apoptosis in the liver. Apoptosis of other liver cells, such as liver sinusoidal endothelial cells in ischemia, lymphocytes in viral hepatitis, and cholangiocytes in PBC and PSC can all contribute to disease pathogenesis [60]. It is important to point out that TNF engagement in hepatocytes usually does not result in cell death due to the simultaneous activation of the NFκB system. Nevertheless, apoptosis has been observed in many animal models of liver disease, and while an inhibition of apoptosis protects rodents from injury such as NASH, clinical trials in humans have failed to prove these inhibitors beneficial in preventing injury or fibrosis [112,113]. Whether this is due to a lack of potency, adequate delivery to target tissues, imbalance between fibrogenesis and fibrinolysis, or pan-caspase inhibitors resulting in a switch in the cell death mode from apoptosis to necroptosis remains to be studied [112]. The occurrence of apoptosis in clinical settings and pathological conditions such as fulminant liver failure (due to AIH, virus, or toxicity) has been called into question, as histologically sub-massive necrosis of hepatocytes with a lytic cell death morphology is reported. In parallel, in animal models such as BDL, some have argued that apoptosis does not occur given the necrotic appearance of areas of bile duct infarct. Due to these observations, many investigators have studied the newer cell death subroutines, necroptosis and pyroptosis, and to some extent ferroptosis, in various liver disease models with often contradictory and confusing results. Different labs have employed a slew of inhibitors with different specificities and concluded that a cell death subroutine is blocked based on the function of the inhibitor [27]. Many of these studies focus on upstream initiators of death pathways such as RIPK1, RIPK3, and NLRP3, which do not always result in cell death but rather modulate inflammation [27,130,131]. The participation of these proteins in other signaling pathways and their “moonlighting” functions have not been adequately studied. For example, we now know that MLKL does not only form pores in the cell membrane to affect lysis but also participates in vesicle export and trafficking by engaging the ESCRTIII machinery [31]. Furthermore, a lack of proper experimental controls, excessive reliance on immunostaining of injured and reactive tissue with non-specific polyclonal antibodies, lack of proper littermate controls and sub-strain matching, and non-standardized protocols for animal disease models (including various diets for the NASH/NAFLD rodent models) all contribute to the confounding results [27].

Lastly, it is important to note that various cell death modes can coexist, and cells can switch between one death subroutine to another. It is well known that there is considerable crosstalk between the various cell death modes and signaling cascades, which has led to the proposal of PANoptosis as a novel cell death routine in which the simultaneous engagement of various cell death modes is observed [277,278,279,280]. PANoptosis is a form of inflammatory cell death during which the major cell death pathways, apoptosis, necroptosis, and pyroptosis, can be activated in parallel. The trigger for this type of cell death can be DAMPS, PAMPS, or viral pathogens activating the ZBP1/DAI intracellular sensor [280]. When engaged, this cell death pathway can lead to the assembly of a large multiprotein complex termed the PANoptosome, which consists of the components of the apoptosis, necroptosis, and pyroptosis pathways [278].

In hepatocytes, the default cell death mode in the presence of intact caspases is still likely apoptosis. However, we must keep an open mind, as many cell death subroutines can be activated with the same signal in hepatocytes. For example, mitochondrial damage leading to excessive ROS can activate MPT necrosis, pyroptosis, and ferroptosis (lipid-ROS). It is the circumstances that dictate which death routine is dominant, or whether there is synergy and the simultaneous activation of multiple pathways in hepatocytes needs to be considered. Furthermore, in addition to hepatocytes, the liver consists of various cell types including parenchymal, cholangiocytes and non-parenchymal resident KCs, NKT cells, liver sinusoidal endothelial cells, and stellate cells, making it possible that different cell death pathways can occur in these diverse cell types. Future studies need to focus on understanding the connection between various cell death pathways and the possible role of PANoptosis in liver disease to improve our understanding of these intricacies and connections in disease pathogenesis and aid in the development of therapeutics.

## Figures and Tables

**Figure 1 ijms-21-09682-f001:**
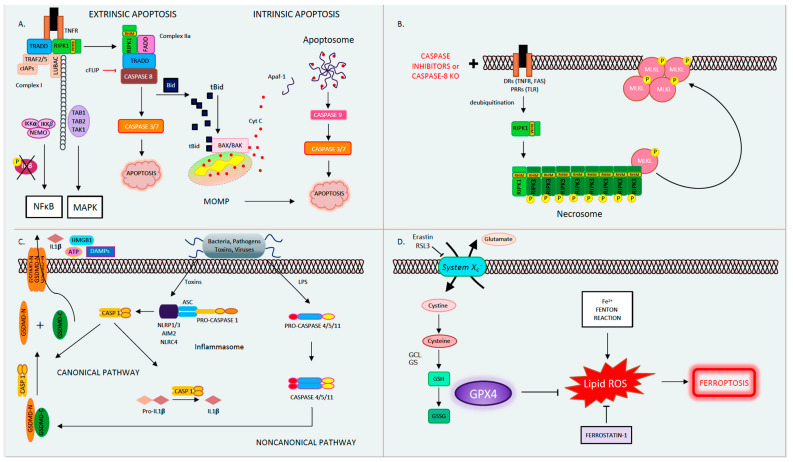
The four different types of regulated cell death. (**A**). Apoptosis. Apoptosis is a form of orderly programmed cell death, occurring both via an extrinsic pathway and an intrinsic pathway. In the extrinsic pathway, after a DR is engaged (TNFR as an example depicted here) by its ligand (here, TNF), it oligomerizes and binds the adaptor TRADD, the E3 ubiquitin ligases TRAF2/5, cIAPs, and LUBAC, as well as the kinase RIPK1 to form Complex I. Then, RIPK1 becomes polyubiquitinated and serves as a scaffolding protein for the further recruitment of TAK1-TAB1/2 and IKKα/β-NEMO. TAK1 activates the MAPK pathway, while phosphorylation and the subsequent degradation of IκB leads to NFκB activation and the transcription of pro-survival/inflammatory genes. If RIPK1 is deubiquitinated, it forms Complex IIa, consisting of FADD, TRADD, RIPK1, and caspase 8. Caspase 8 becomes activated in Complex IIa, in turn activating its downstream executioner caspases 3 and 7, leading to apoptosis. In the intrinsic pathway, caspase 8 mediates the cleavage of Bid into tBid. tBid translocates to the mitochondria where it binds BAX/BAK, leading to MOMP and Cytochrome C release. Cytochrome C binds Apaf-1, releasing its auto-inhibitory hold and allowing for its oligomerization, apoptosome formation, and caspase 9 activation. In turn, caspase 9 activates the executioner caspases 3 and 7, leading to apoptosis. (**B**). Necroptosis occurs when death receptors (such as TNF and FAS) or pattern recognition receptors (such as TLR2/3) are activated in the presence of pan-caspase inhibitors to block apoptosis. In this instance, deubiquitinated RIPK1 forms Complex IIb (also known as the necrosome) by associating with RIPK3 through their shared RIP homology interaction motif (RHIM) to form the necrosome. RIPK3 auto-phosphorylates, polymerizes, and in turn recruits and phospho-activates MLKL to the necrosome (pMLKL). Subsequently, pMLKL translocates to the plasma membrane where it forms a tetramer and permeabilizes the cell membrane causing oncolysis. (**C**). Pyroptosis. Pyroptosis occurs through two pathways. In the canonical pathway, upon the recognition of external toxins or pathogens, the supramolecular inflammasome (consisting of NLRPs, NLRC4, AIM2, the adaptor ASC, and pro-caspase 1) forms. Pro-caspase 1 undergoes autocatalytic cleavage to form activated caspase 1. In turn, caspase 1 cleaves pro-IL-1β into active IL-1β and GSDMD into GSDMD-C and GSDMD-N. GSDMD-N translocates to the plasma membrane where it forms a pore, releasing DAMPs (such as ATP and HMGB1) and IL-1β. In the noncanonical pathway, the detection of LPS directly activates caspases 4/5/11, which then cleaves GSDMD to form the membrane pore and induce pyroptosis. (**D**). Ferroptosis. Ferroptosis is a type of regulated cell death that is driven by intracellular lipid peroxidation and is highly dependent on ROS generation and iron availability. Cystine enters the cell by the cystine/glutamate antiporter (*System X_c_^-^*). Within the cell, cystine is reduced to cysteine, and glutamate cysteine ligase (GCL) catalyzes the synthesis of γ-glutamyl cysteine from glutamate and cysteine. Glutathione synthase (GS) generates glutathione (GSH) by adding glycine. GSH reduces free radicals when it is reduced to GSSG by glutathione peroxidase-4 (GPX4). GPX4 and other free radical scavengers such a ferrostatin-1 inhibit lipid peroxidation (lipid-ROS). In the absence of GPX4, lipid peroxides accumulate driving ferroptosis. **Abbreviations:** AIM2: absent in melanoma-2, APAF1: apoptotic peptidase activating factor 1, ASC: apoptosis-associated speck-like protein containing a CARD, ATP: adenosine triphosphate, BAK: BCL2 antagonist/killer, BAX: BCL-2-like protein 4, CASP: caspase, cIAPs: cellular inhibitor of apoptosis, DAMP: danger-associated molecular patterns, DR: death receptor, FADD: Fas associated via death domain, GCL: glutamate cysteine ligase, GPX4: glutathione peroxidase-4, GS: glutathione synthase, GSDMD: gasdermin D, IL: interleukin, HMGB1: high mobility group box 1, IKKα/β: IKB kinase, LPS: lipopolysaccharide, LUBAC: linear ubiquitin chain assembly complex, MAPK: mitogen-activated protein kinase, MLKL: pseudokinase mixed lineage domain-like, MOMP: mitochondrial outer membrane pore formation, NEMO: NF kappa B essential modulator, NLRC: NLR family CARD domain-containing protein, NLRP: NOD-like receptor family, pyrin domain-containing, RHIM: RIP homology interaction motif, RIPK1/3: receptor-interacting protein kinase 1/3, ROS: reactive oxygen species, TAB1/2: TAK binding protein, TAK1: transforming growth factor-activated kinase-1, TLR2/3: Toll-like receptors 2/3, TNF: tumor necrosis factor, TNFR: tumor Necrosis Factor Receptor, TRADD: TNFR-associated death domain, TRAF2/5: TNF receptor-associated factors 2/5.

**Figure 2 ijms-21-09682-f002:**
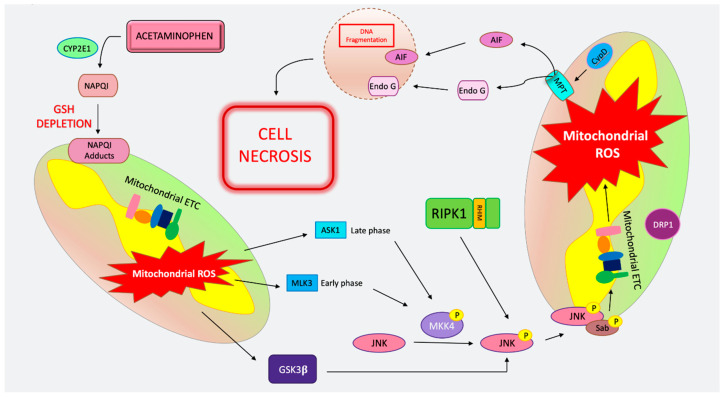
Mechanism of acetaminophen-induced cell death. During conditions of acetaminophen (APAP) overdose, when sulfation and glucuronidation are saturated, excessive APAP is converted by the CYP450 enzymes (mainly CYP2E1) to its reactive metabolite N-acetyl-p-benzoquinone imine (NAPQI). NAPQI is inactivated by conjugating with GSH. However, excessive amounts of NAPQI generated by APAP overdose deplete cytoplasmic and mitochondrial GSH and bind to protein thiols and cysteine residues to form NAPQI adducts leading to ER stress, mitochondrial oxidative dysfunction, and an accumulation of ROS. ROS accumulation activates the mitogen-activated protein kinase (MAPK) cascade. MAPKs, MLK3, and ASK1 go on to phosphorylate MKK4, which in turn phospho-activates c-Jun-N-terminal Kinase (JNK). GSK3β and RIPK1 have also been shown to be able to activate pJNK. Then, activated JNK (pJNK) translocates to the mitochondrial membrane, where it binds to and phosphorylates Sab, causing the inhibition of the mitochondrial electron transport chain (ETC), further accumulation of ROS and mitochondrial dysfunction, which eventually lead to cyclophilin D-mediated MPT, mitochondrial membrane potential collapse, and the release of apoptosis-inducing factor (AIF) and endonuclease G into the cytosol. Endonuclease G and AIF translocate to the nucleus, causing DNA fragmentation resulting in cellular collapse and necrosis. **Abbreviations:** AIF: apoptosis-inducing factor, APAP: acetaminophen, ETC: electron transport chain, GSH: glutathione, GSK3β: glycogen synthase kinase 3 beta, JNK: cJun-N-terminal, MAPK: mitogen activated protein kinase, MAPKKK: mitogen activated protein kinase kinase kinase, MKK: mitogen activated protein kinase, MPT: mitochondrial permeability transition, NAPQI: N-acetyl-*p-*benzoquinone imine, pJNK: phosphorylated cJun-N-terminal, RIPK1: receptor-interacting protein kinase 1, ROS: reactive oxygen species.

**Table 1 ijms-21-09682-t001:** Main findings of cell death modes in liver diseases.

	Apoptosis	MPT-Mediated Necrosis	Necroptosis	Autophagy	Pyroptosis	Ferroptosis
Alcoholic Liver Diseases (ALD)	-Extensively studied in human and murine models. -ROS production and CHOP-dependent apoptosis [61,62,63,64]. -FAS, FASL, and TNF well characterized [66,67]. -Casp8^Δhepa^ mice had less steatosis and cell death [72]. -Pan-caspase inhibitor protects and no switch to necroptosis [72,73]. -BID KO mice protected from apoptosis but not inflammation [74]		-Higher RIPK3, but the source is not clear [30,76]. -RIPK3 global KO mice treated with alcohol had less steatosis, inflammation, and liver injury [76]. -Global RIPK3 KO less transaminases but not difference in inflammation and neutrophil infiltration [77]. -Daily 7-Nec1 reduced inflammation but did not prevent liver injury [77]. -No transcriptional induction of RIPK3 but proteasome inhibition resulted in increase [77]. -No induction of RIPK3 in Gao-Binge model. -Further studies needed to explore the role of pMLKL. Unconfirmed.	-Interplay of autophagy and apoptosis is highly likely in pathogenesis. -Ethanol induces autophagy through ROS and activation of AAMPK and inhibition of mTORC1 [79,82]. -Dose and duration of alcohol can influence autophagy with acute and lower dose models, promoting and higher doses inhibiting some components and preventing autophagy [79]. -Parkin-mediated mitophagy and CMA were shown to be protective [85,87].	-More prominent in inflammatory cells than hepatocytes [43,89,90]. -NLRP3, CASP-1, and ASC upregulated with ethanol [91]. -Alcohol induces CASP-1, NLRP3, IL-1. CASP-1 KO, and IL-1 antagonist protection [92]. -Elevated CASP11/4 activate GSDMD, more prominent in macrophages [96].	-Several studies suggest, however, that GPX4 may not be the initiator [57]. -SIRT1 deficiency may be protective [98]. -Lipin1 overexpression (lipid metabolism) worsens injury, increased iron accumulation, impaired ferroptotic gene expression, GPX4 not altered [57].
Non-Alcoholic Fatty Liver Diseases (NASH/NAFLD)	-Patients with NASH have positive detection of CASP3/7 and TUNEL [99]. -CASP3/8 KO mice on MCD diet protected from apoptosis [105,106]. -Emricasan protects mice from HFD [107]. Caspase inhibitors not successful in human clinical trials [111,112,113]. -Conflicting results of pan-caspase inhibition in different diets [108,109]. -CASP6 interplay with CASP3/7 leads to persistent apoptosis involving the AMPK pathway [114]. -Both intrinsic (via lipotoxocity and organelle stress) and extrinsic (via cell surface receptor) pathways contribute to injury [117,118,119,120].	-CypD KO mice have lower mitochondrial stress, steatosis, and TG on HFD [16]. -Cyclophilin inhibitor reduced inflammation, steatosis, and ballooning in mice on HFD [121].	-Increased RIPK3 in humans with NASH [27,122,123]. -RIPK3 induction with MCD diet further exacerbated with CASP8 deletion [123]. -Increased RIPK3 and pMLKL in HFD mice [124]. -RIPK3 KO mice shown to be glucose intolerant on chow diet [124,125] and had worse injury [124]. -RIPK3 deficiency leads to less inflammation, steatosis, and fibrosis, in CD-HFD and MCD diet [126]. -MLKL KO mice on HFD had increased MLKL, RIPK1, and pMLKL. [127]. -RIPK1 inhibitor prevented inflammation and decreased fibrosis in HFD mice [128]. -Autophagic markers were abrogated by MLKL KO, suggesting that MLKL may play a role in autophagic flux [129]. -Occurrence of necroptosis under basal conditions without pan-caspase inhibition is questionable.		-In NASH inflammasome activation is triggered by lipotoxicity, organelle stress, and hepatocyte death [133]. -Inflammasome-mediated dysbiosis has been shown to regulate NAFLD and obesity in MCD diet [134]. -CASP1 KO, ASC KO, or NLRP3 KO mice on MCD diet showed increased transaminases, worsening inflammation, and steatosis compared to WT controls. More injury was evident in IL-18 KO mice but not in IL-1R KO mice [134]. -NLRP3 inhibitor, in *foz/foz* mice fed HFD and WT mice with MCD diet improved inflammation, but there was no effect on steatosis [135]. -GSDMD expression was increased in db/db mice on MCD diet, and GSDMD KO mice on HFD had attenuated steatosis, improved ALT and TG levels, and less fibrosis [47].	-Vit E and phlebotomy improves liver chemistry in NAFLD/NASH [138,139] -GPX4 induction has been observed following Western and MCD diets [140]. -GPX4 activator administration promoted cell survival and improved AST/ALT [140]. -Trolox (Vit E analogue and antioxidant) led to less cell death [141]. -Ferrostatin treatment reversed mitochondrial morphological changes observed with MCD diet [142]. -Distinguishing ROS leading to MPT and necrosis from ferroptosis is still difficult.
Acetaminophen (APAP) toxicity	-APAP cell death is not morphologically apoptotic. -Translocation of tBID to mitochondria during APAP does not mediate MOMP [170,174]. -Caspase inhibitors do not protect from APAP toxicity [171,174]. -No evidence MOMP in APAP toxicity [155,161]. -TUNEL staining is not specific to apoptosis and the pattern of the TUNEL stain in APAP is cytoplasmic not nuclear [177]. -Lack of PUMA protects against hepatocyte death but through preventing necrosis not apoptosis [167]. RIPK1 and JNK inhibition and siRNA knockdown abrogates PUMA’s upregulation [167].	-CypD-deficient mice are protected against APAP hepatotoxicity and cell death [15]. -Cyclosporin protects from APAP [163,164]. -Inhibition, knockdown or KO of MAPK proteins, including MLKL3, ASK1, MKK4, and JNK protects against necrotic cell death [155,156,157,158,159,160]. -AIF-deficient mice have less injury in APAP toxicity [162]. -Knockdown or inhibition of GSK3β and RIPK1 protects from APAP liver injury [30,166,167]. -RIPK1 participates upstream, of JNK and has a non-necroptotic function [30,167].	-Nec-1 has been shown to protect from APAP; however, it has off-target effects [27,179,180]. -Controversial results with global RIPK3 KO mice treated with APAP have been reported [27,30,165,181]. -MLKL KO mice not protected from APAP [30,36]. -RIPK1 may have a role in APAP toxicity independent of its function in necroptosis and upstream of JNK signaling [30,167].	-Activation of autophagy protective in APAP [182,183,184,185]. -Autophagy and mitophagy limit ROS generation and are thought to limit the expansion of necrotic foci [188]. -LC3 transgenic mice show an increase in autophagosomes surrounding necrotic foci after APAP administration [152,188]. -APAP increases autophagic flux and its inhibition increases APAP protein adducts [152].	-CASP1, ASC, and NLRP3 KO mice have been shown to be less sensitive to APAP toxicity [189]. -No protection with anti-IL-1β antibodies or KO of IL-1β, CASP1 or NALP3 [190,191,192]. - TLR4 signaling in macrophages proposed to aggravate APAP injury by generating IL-1α [192]. However, this is controversial. -GSDMD KO mice have increased injury in APAP toxicity [194]. -Further investigation is needed to determine pyroptosis in APAP toxicity.	-Antioxidants such as Fer-1 and deferoxamine may alleviate APAP toxicity with less lipid peroxidation, GSH depletion, and iron accumulation [198,199,200]. -No cleaved caspase 3 and increased RIPK3 expression was detected with Fer-1 treatment [201]. -DFO protected against APAP by maintaining GSH levels [201]. -It is difficult to determine if antioxidants work to increase GSH and prevent APAP necrosis or if they are preventing “ferroptosis”. -No data on GPX4 KO mice are available; therefore, the contribution of ferroptosis is controversial.
Autoimmune Hepatitis (AIH) including mouse models of Concanavalin A (ConA) and α-galactoceramide (α-GalCer) models	-Councilman bodies that are apoptotic have been seen in liver biopsy of AIH patients [203]. -In the ConA model IFN-γ, IL-4, IL-6, FAS, and TRAIL participation has been reported, but TNF is known as the principal cytokine involved in toxicity [206]. -TNF induces FasL-dependent apoptosis with transcriptional induction of FAS via the NFκB pathway [27,207]. -Caspase inhibition in ConA model is largely ineffective [209].		-Increased hepatic MLKL mRNA and protein expression in the liver biopsy of patients with AIH [36]. -Nec-1 has been shown to protect against ConA [201,202,203,204,205,206,207,208,209,210,211,212]. -MLKL global KO has protected from Con but MLKL-hepatocyte-specific KO is not protected [36,214]. -RIPK1 inhibition has shown to be effective upstream of MLKL in a ConA model [36]. -RIPK1-deficient hepatocytes die of TNF and caspase-dependent apoptosis with low dose ConA injection [215,216]. -RIPK1 knockdown promoted increased TNF-mediated apoptosis in αGalCer [205]. -RIPK3 deficiency in αGalCer controversial. One paper has shown a modest protective effect by limiting cytokine production and NK activation [217]. In other, RIPK3 global KO and MLKL global KO mice not protected from αGalCer [205].		-NLRP3 and CASP1 KO mice were protected in a ConA model [219]. -CASP1 is needed for IL-1β production [219]. -IL-1 suppression reduces liver injury through reduced TNF and IL-17 secretion in a ConA model [219]. -NKT cells express OX40 and high levels of CASP1. The activation of CASP1 led to pro–IL-1β maturation and GSDMD cleavage and pyroptosis [221]. -Low-dose ConA induces TNF superfamily receptor, OX40, in the liver and blocking OX40 can prevent ConA hepatitis [221]. -Further studies on GSDMD are needed to study Pyroptosis in ConA and αGalCer.	-IDO-dependent ferroptosis has been reported in the ConA model [222]. -Downregulation of Cav-1 increases RNS production and injury is dampened with Fer-1 administration [223]. -No data with GPX4; therefore, results are inconclusive.

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
