# Peer review of "Cell Death in Liver Diseases: A Review"

_ijms, 2020, doi:10.3390/ijms21249682_

Round 1

Reviewer 1 Report

In this review, the authors described the effects of regulated cell death in liver diseases. The review comprehensively covers multiples types of regulated cell death in various non-malignant liver diseases. Overall, I think it is a well-written review. Some minor points I think would help to deliver the message in a better way if you can make a table that include the previous research findings so that people can easily find the information they need. 

Author Response

Response to reviewer 1 comments

Thank you for the excellent suggestion. We added a table in Part 8 – discussion (Page 28) that includes previous research and main findings of each cell death pathway in the main liver diseases discussed.

We also critically edited the manuscript and fixed a number of errors and typos.

Enclosed you will see two files: one with changes tracked in red for your review, and one final version with changes accepted.

Reviewer 2 Report

Authors provide a review of cell death in various liver diseases. Overall, the review is well written and informative. Please see the comments below. 

  1. There are several grammatical errors, large spaces, and bolded and / or underlined sentences that require attention. 
  2. There is an error in the sentence in lines 84-85. Authors need to address where it says "was not, was even more". 
  3. Authors should expand on autophagy-induced cell death. This topic is actually controversial. Many believe autophagy to be a protective pathway and not a cell-death inducing pathway. 
  4. Authors should expand on autophagy in ALD. Autophagy is thought to be activated by acute alcohol treatment, but many have shown that it is actually blocked by chronic alcohol treatment. 

Author Response

Response to reviewer 2 comments

Thank you for your valuable comments. We have applied the changes requested to enhance the quality of our review.

Point 1: There are several grammatical errors, large spaces, and bolded and / or underlined sentences that require attention. 

The reviewer is correct, looking back the final version of our manuscript did not contain these font anomalies. Something must have occurred when we uploaded the file to the journal. However, it was our error for not double checking the uploaded file and we apologize for the inconvenience. We have addressed font issues and edited the entire manuscript to make sure there are no further grammatical errors or typos. Enclosed you will see two files: one with changes tracked in red for your review, and one final version with changes accepted.

Point 2: There is an error in the sentence in lines 84-85. Authors need to address where it says "was not, was even more". 

Thank you for pointing this out. We have corrected the statement.

Point 3: Authors should expand on autophagy-induced cell death. This topic is actually controversial. Many believe autophagy to be a protective pathway and not a cell-death inducing pathway.

Thank you for the comment. We expanded autophagy and autophagy- dependent cell death in the introduction part 1.4.

Point 4: Authors should expand on autophagy in ALD. Autophagy is thought to be activated by acute alcohol treatment, but many have shown that it is actually blocked by chronic alcohol treatment. 

Thank you for the suggestion, this is an important point we had missed. We expanded the autophagy in ALD section and addressed the effect of ethanol on autophagy in more detail in section 2.3.